Manuscript prepared for Atmos. Chem. Phys.
with version 2015/04/24 7.83 Copernicus papers of the LATEX class copernicus.cls.
Date: 28 July 2016

# The Tropical Tropopause Inversion Layer: Variability and Modulation by Equatorial Waves

Robin Pilch Kedzierski[1], Katja Matthes[1,2], and Karl Bumke[1]

[1]Marine Meteorology Department, GEOMAR Helmholtz Centre for Ocean Research Kiel, Kiel, Germany.
[2]Faculty of Mathematics and Natural Sciences, Christian-Albrechts-Universität zu Kiel, Kiel, Germany.

*Correspondence to:* Robin Pilch Kedzierski (rpilch@geomar.de)

**Abstract.**

The Tropical Tropopause Layer (TTL) acts as a 'transition' layer between the troposphere and the stratosphere over several kilometers, where air has both tropospheric and stratospheric properties. Within this region, a fine-scale feature is located: the Tropopause Inversion Layer (TIL), which

consists of a sharp temperature inversion at the tropopause and the corresponding high static stability values right above, which theoretically affect the dispersion relations of atmospheric waves like Rossby or Inertia-Gravity waves and hamper stratosphere-troposphere exchange (STE). Therefore, the TIL receives increasing attention from the scientific community, mainly in the extratropics so far. Our goal is to give a detailed picture of the properties, variability and forcings of the tropical TIL,

with special emphasis on small-scale equatorial waves and the QBO.

We use high-resolution temperature profiles from the COSMIC satellite mission, i.e. ∼2000 measurements per day globally, between 2007 and 2013, to derive TIL properties and to study the fine-scale structures of static stability in the tropics. The situation at near tropopause level is described by the 100hPa horizontal wind divergence fields, and the vertical structure of the QBO is provided

by the equatorial winds at all levels, both from the ERA-Interim reanalysis.

We describe a new feature of the equatorial static stability profile: a secondary stability maximum below the zero wind line within the easterly QBO wind regime at about 20-25km altitude, which is forced by the descending westerly QBO phase and gives a double-TIL-like structure. In the lowermost stratosphere, the TIL is stronger with westerly winds. We provide the first evidence

of a relationship between the tropical TIL strength and near-tropopause divergence, with stronger (weaker) TIL with near-tropopause divergent (convergent) flow, a relationship analogous to that of TIL strength with relative vorticity in the extratropics.

To elucidate possible enhancing mechanisms of the tropical TIL, we quantify the signature of the different equatorial waves on the vertical structure of static stability in the tropics. All waves show,

on average, maximum cold anomalies at the thermal tropopause, warm anomalies above, and a net TIL enhancement close to the tropopause. The main drivers are Kelvin, inertia-gravity and Rossby

waves. We suggest that a similar wave modulation will exist at mid and polar latitudes from the extratropical wave modes.

## 1 Introduction

The Tropopause Inversion Layer (TIL) is a narrow region characterized by temperature inversion and enhanced static stability located right above the tropopause. This fine-scale feature was discovered by tropopause-based averaging of high-resolution radiosonde measurements by Birner et al. (2002) and Birner (2006). Satellite Global Positioning System radio occultation observations (GPS-RO) show that the TIL is present globally (Grise et al., 2010).

Static stability is a parameter used in a number of wave theory approximations, thus affecting the dispersion relations of atmospheric waves like Rossby or Inertia-Gravity waves (Birner, 2006; Grise et al., 2010). Also, static stability suppresses vertical motion and correlates with sharper trace gas gradients, inhibiting the cross-tropopause exchange of chemical compounds (Hegglin et al., 2009; Kunz et al., 2009; Schmidt et al., 2010). For these reasons, the TIL attracts increasing interest from 40 the scientific community.

There is a considerable body of research about the TIL in the extratropics, establishing the TIL as an important feature of the extratropical upper-troposphere and lower-stratosphere (Gettelman et al., 2011). In the tropics, the transition between the troposphere and the stratosphere is considered to happen over several kilometers, dynamically and chemically (Fueglistaler et al., 2009; Gettelman 45 and Birner, 2007), but less is known about the tropical TIL, as the following review will show.

In the extratropics, climatological studies have shown that the TIL reaches maximum strength during polar summer (Birner, 2006; Randel et al., 2007; Randel and Wu, 2010; Grise et al., 2010), whereas the TIL within anticyclones in mid-latitude winter is of the same strength or even higher from a synoptic-scale point of view (Pilch Kedzierski et al., 2015). Several mechanisms for ex-50 tratropical TIL formation/maintenance have been studied: water vapor radiative cooling below the tropopause (Randel et al., 2007; Hegglin et al., 2009; Kunz et al., 2009; Randel and Wu, 2010), dynamical heating above the tropopause from the downwelling branch of the stratospheric residual circulation (Birner, 2010), tropopause lifting and sharpening by baroclinic waves and their embedded cyclones-anticyclones (Wirth, 2003, 2004; Wirth and Szabo, 2007; Son and Polvani, 2007; Randel 55 et al., 2007; Randel and Wu, 2010; Erler and Wirth, 2011), and the presence of small-scale gravity waves (Kunkel et al., 2014). A high-resolution model study by Miyazaki et al. (2010a, b) suggests that radiative effects dominate TIL enhancement in polar summer whereas dynamics are the main drivers in the remaining latitudes and seasons.

On the other hand, very little research has focused on the tropical TIL. Bell and Geller (2008) 60 showed the TIL from one tropical radiosonde station, and Wang et al. (2013) reported a slight weakening of the tropical TIL between 2001-2011. Grise et al. (2010) included the horizontal and vertical

variability of the tropical TIL in their global study about near-tropopause static stability, which is so far the most detailed description of the TIL in the tropics. They found the strongest TIL centered at the equator in the layer 0-1km above the tropopause, peaking during NH winter. This agrees well with the seasonality and location of tropopause sharpness as described later by Son et al. (2011) and Kim and Son (2012). The horizontal structures in seasonal mean TIL in the tropics are reminiscent of the equatorial stationary wave response associated with climatological deep convection (Grise et al., 2010; Kim and Son, 2012). Grise et al. (2010) also noted that static stability is enhanced in the layer 1-3km above the tropopause during the easterly phase of the quasi-biennial oscillation (QBO).

Equatorial waves influence the intraseasonal and short-term variability of the temperature structure near the tropical tropopause (Fueglistaler et al., 2009). Kelvin waves and the Madden-Julian oscillation (MJO) (Madden and Julian, 1994) were reported as the dominant modes of temperature variability at the tropopause region (Kim and Son, 2012). Equatorial waves cool the tropopause region (Grise and Thompson, 2013), and also produce a warming effect above it (Kim and Son, 2012). This wave effect forms a dipole that can sharpen the gradients that lead to TIL enhancement, but no study has quantified this effect so far.

Our study aims to describe the tropical TIL, its variability and forcings in detail, in order to increase the knowledge about its properties and highlight this sharp and fine-scale feature within the tropical transition layer between the troposphere and the stratosphere. Section 2 will show the datasets and methods used in our analyses. Section 3 will describe the vertical and horizontal structure and day-to-day variability of the TIL, its relationship with near-tropopause divergence, and the influence of the QBO on the vertical structure of static stability and TIL strength in particular. In section 4 we quantify the signature of the different equatorial waves and their effect on the mean temperature and static stability profiles in tropopause-based coordinates. Section 5 will discuss the applicability of our results with equatorial wave modulation to the extratropical TIL, given the different wave spectrum in the extratropics, and section 6 sums up the results.

## 2   Data and Methods

### 2.1   Datasets

We analyze temperature profiles from GPS radio occultation (GPS-RO) measurements which are provided at a 100m vertical resolution, from the surface up to 40km altitude, comparable to high-resolution radiosonde data. Although the effective physical resolution of GPS-RO retrievals is of ∼1km, it improves in regions where the stratification of the atmosphere changes, such as the tropopause and the top of the boundary layer, i.e. the vertical resolution is highest where it is most needed (Kursinski et al., 1997). The advantage of GPS-RO is based on its global coverage, high sampling density of ∼2000 profiles/day, and weather-independence. We mainly use data from the COSMIC satellite mission (Anthes et al., 2008) for the years 2007-2013. For Figure 1 only, we added two ear-

lier GPS-RO satellite missions: CHAMP (Wickert et al., 2001) and GRACE (Beyerle et al., 2005), which provide less observations (around 200 profiles/day together) for 2002-2007.

The situation at near-tropopause level is retrieved from the ERA-Interim reanalysis (Dee et al., 2011). We make use of horizontal wind divergence and geopotential height fields at 100hPa on a 2.5°×2.5° longitude-latitude grid and 6-hourly time resolution, and also daily-mean vertical profiles of the zonal wind at the equator, for the time period 2007-2013. We choose the 100hPa level because it is the standard pressure level from ERA-Interim that is closest to the climatological tropopause in the tropics (96-100hPa in summer, 86-88hPa in winter, Kim and Son (2012)). Tropical winds near the tropopause in ERA-Interim differ from observations slightly more than in the extratropics, but still are of good quality (Poli et al., 2010; Dee et al., 2011), and the variability of horizontal divergence is in balance with temperatures, which are constrained by GPS-RO in the UTLS.

### 2.2 TIL Strength Calculation

We define the tropopause height ($TP_z$) using the WMO lapse-rate tropopause criterion (WMO, 1957). Given the strong negative lapse-rate found above the tropopause near the equator, the lapse-rate tropopause nearly coincides with the cold-point tropopause most of the time. This is in agreement with earlier studies, which did not find substantial differences in their results applying different tropopause definitions (Grise et al., 2010; Wang et al., 2013). From the GPS-RO temperature profiles, vertical profiles of static stability are calculated as the Brunt-Väisälä frequency squared ($N^2$ [$s^{-2}$]):

$$N^2 = (g/\Theta) \cdot (\partial\Theta/\partial z)$$

where g is the gravitational acceleration, and $\Theta$ the potential temperature. Profiles with unphysical temperatures or $N^2$ values (temperature <-150°C, >150°C or $N^2 > 100 \times 10^{-4} s^{-2}$) and those where the tropopause cannot be found have been excluded. TIL strength (sTIL) is calculated as the maximum static stability value ($N^2_{max}$) above the tropopause level. This sTIL measure is commonly used (Birner et al., 2006; Wirth and Szabo, 2007; Erler and Wirth, 2011; Pilch Kedzierski et al., 2015), because it makes sTIL independent of its distance from the tropopause and $N^2$ is a physically relevant quantity. Our algorithm searches for $N^2_{max}$ in the first 3km above $TP_z$, but most often finds it in the first kilometer.

### 2.3 Mapping of TIL Snapshots

The procedure to derive daily TIL snapshots in this study is similar to the method by Pilch Kedzierski et al. (2015), but with a longitude-latitude projection.

The daily TIL snapshots were estimated at a 5° longitude-latitude grid between 30°S-30°N. For each grid point we calculate the mean $N^2_{max}$ from all GPS-RO profiles within +-12.5° longitude-

130 latitude to account for the lower GPS-RO observation density in the tropics compared to the extrat-ropics (Son et al., 2011). This setting avoids gaps appearing in the maps, and smooths out undesired small-scale features that cannot be captured with the current GPS-RO sampling.

We also produce similar maps of 100hPa horizontal wind divergence. For a fair comparison with the TIL snapshots, we equal the spatial scale and follow the same method, but instead of averaging 135 $N^2_{max}$ values, we use the collocated divergence of each GPS-RO profile: the value from the nearest ERA-Interim grid point and 6h time period to each observation. Examples of TIL snapshots can be found in Figure 2 (section 3.1.2). If plotted at full horizontal resolution, divergence would show small-scale features superimposed over the synoptic-to-large scale structures in Fig. 2, making the comparison with $N^2_{max}$ more difficult.

**2.4   Wavenumber-Frequency Domain Filtering**

Our purpose is to extract the temperature and $N^2$ signature of the different equatorial wave types on the zonal mean vertical profiles. For this, we follow Wheeler and Kiladis (1999), that studied equatorial wave signatures on the outgoing longwave radiation (OLR) spectrum observed from satellites by wavenumber-frequency domain filtering. Wheeler and Kiladis (1999) give an in-depth description 145 of the theoretical and mathematical background of the filtering methods.

Theoretically, the equatorial wave modes are the zonally and vertically propagating, equatorially trapped solutions of the 'shallow water' equations (Matsuno, 1966; Lindzen, 1967) characterized by four parameters: meridional mode number (*n*), frequency (*v*), zonal planetary wavenumber (*s*) and equivalent depth (*h*). Each wave type (Kelvin, Rossby, the different modes of Inertia-Gravity waves) 150 has a unique dispersion curve in the wavenumber-frequency domain, given its mode *n* and equivalent depth *h*.

Wheeler and Kiladis (1999) found that the equatorial OLR power had spectral signatures that were significantly above the background. The signature's regions in the wavenumber-frequency domain match with the dispersion curves of the different equatorial wave types. Also, they found signatures 155 outside of the theoretical wave dispersion curves that have characteristics of the Madden-Julian oscillation (MJO) (Madden and Julian, 1994).

Our method is similar to that of Wheeler and Kiladis (1999), but analyzes temperature and $N^2$ at all levels between 10-35km altitude instead of OLR. For filtering, the data must be periodic in longitude and time and cover all longitudes of the equatorial latitude band. Therefore, the COSMIC 160 GPS-RO profiles (Anthes et al., 2008) need to be put on a regular longitude grid on a daily basis. We explain how this is done in the follwing section (2.4.1). More details about our proceeding with the filter, and the differences from Wheeler and Kiladis (1999) can be found in section 2.4.2. Note that this method is only used in Figs. 5 and 6 (section 4).

### 2.4.1 Gridding of GPS-RO profiles

The COSMIC GPS-RO temperature profiles between 10°S-10°N are gridded daily on a regular longitude grid with a 10° separation. At each grid point, the profiles of that day within 10°S-10°N and +-5° longitude are selected to calculate a tropopause-based weighted average temperature profile and the corresponding $N^2$ vertical profile:

$$T_{grid}(\lambda, Z_{TP}, t) = \sum_i w_i T_i(\lambda, Z_{TP}, t)/\sum_i w_i$$

$$N^2_{grid}(\lambda, Z_{TP}, t) = \sum_i w_i N^2_i(\lambda, Z_{TP}, t)/\sum_i w_i$$

where $\lambda$ is longitude, $Z_{TP}$ is the height relative to the tropopause and t is time. The weight $w_i$ is a Gaussian-shape function that depends on the distance of the GPS-RO profile from the grid center, taking longitude, latitude and time (distance from 12UTC):

$w_i = exp(-[(D_x/5)^2 + (D_y/10)^2 + (D_t/12)^2])$ , where D are the distances in °longitude (x sub-

175 script), °latitude (y) and hours (t). The maximum distance allowed from the grid point in each dimension is: 5° longitude, 10° latitude, and 12 hours from 12UTC, respectively.

The gridded tropopause height ($\lambda$,t) is calculated with the same weighting of all profiles' tropopauses. The gridded temperature and $N^2$ profiles are shifted, as the last step, from the tropopause-based vertical scale onto a ground-based vertical scale from 10km to 35km altitude, obtaining a longitude-

180 height array for each day for 2007-2013.

Most often 2-3 profiles are selected for averaging at a grid point with these settings, although one GPS-RO profile is sufficient to estimate a grid point. However, in 6.5% of the cases the algorithm does not find any profile. To fill in the gaps, the longitude range to select the profiles is incremented to +-10° instead of +-5°, which still leaves a 0.8% of empty grid-points. For this minority, profiles are

185 selected within +-1day and +-15° longitude. In all cases the weighting function remains the same. These exceptions are for a very small portion of the gridded data, and therefore do not affect the retrieved wave signatures after filtering.

Our method is essentially an update from Randel and Wu (2005). It is adapted for the higher number of GPS-RO retrievals of the COSMIC mission compared to its predecessors CHAMP and

190 GRACE: Randel and Wu (2005) used a 30° longitude grid and selected profiles for +-2days, while we use 10° spacing and profiles of the same day. This leads to increased zonal resolution, as well as a minimized temporal smoothing. Another difference is that we do the averaging in tropopause-based coordinates to avoid smoothing the TIL, and also use latitude differences in the weighting function. Compared to earlier studies that filtered equatorial waves using GPS-RO data (Randel and Wu, 2005;

Kim and Son, 2012), our daily fields with barely any running mean allow the analysis of waves with

higher frequencies and wavenumbers (i.e. a wide part of the inertia-gravity wave spectrum) that otherwise are smoothed out and not accounted for.

### 2.4.2 Filter Settings

With the longitude-height-time array of gridded temperature and $N^2$ profiles obtained according to section 2.4.1, we proceed with the filtering in the wavenumber-frequency domain as follows. For each vertical level (from 10km to 35km height with 0.1km vertical spacing), a longitude-time array is retrieved, detrended and tapered in time. Then, a space-time bandpass filter is applied using a two-dimensional Fast Fourier Transform. This is done using the freely available 'kf-filter' NCL function (Schreck, 2009).

The bandpass filter bounds in the wavenumber-frequency domain are defined for the following wave types: Kelvin waves, Rossby waves, and all modes of Inertia-Gravity waves: $n$ = (0,1,2). We separate westward-propagating (negative wavenumber $s$) Inertia-Gravity waves ($WIG_n$), the eastward-propagating ones (positive $s$, $EIG_n$), and the zonal wavenumber zero ($_{s=0}IG_n$) in the analysis. Our $WIG_n$ category includes the Mixed Rossby-Gravity wave modes for $n$ = 0. Including the MJO band (which does not belong to any theoretical dispersion curve), we end up with 12 different bandpass filters that are applied to the longitude-time array of gridded temperature and $N^2$, at each vertical level separately. The exact filter bounds are listed in Table 1. They are similar to the ones used by Wheeler and Kiladis (1999) and take into account the faster and not convectively coupled Kelvin waves found by Kim and Son (2012) in the tropopause temperature variability spectrum. We also allow faster $WIG_0$ and Rossby waves in the corresponding filters. Note that the filter bounds defined in Table 1 never overlap in the wavenumber-frequency domain. After filtering, we obtain a daily longitude-height section with the 12 waves' temperature and $N^2$ signatures. We stress that the temperature and $N^2$ fields are filtered independently at each vertical level.

## 3 Structure and Variability of the Tropical TIL

### 3.1 Vertical and Horizontal Structures

#### 3.1.1 Temporal variability of the vertical $N^2$ profile

We first focus on the variation of the equatorial $N^2$ profiles over time. Figure 1 shows the daily evolution of the equatorial (5°S-5°N) zonal mean $N^2$ profile between 2002-2013, with zonal wind contours superimposed (black westerlies, and dashed easterlies), and a grey tropopause line. The years 2002-2006 appear noisier because the number of observations from the CHAMP (Wickert et al., 2001) and GRACE (Beyerle et al., 2005) satellite missions is about 10 times less than the amount of profiles from COSMIC (Anthes et al., 2008) used between 2007-2013. Therefore local anomalies have a bigger impact on the zonal mean vertical profile during 2002-2006.

The tropopause height in Fig. 1 has a seasonal cycle with a generally higher (lower) tropopause and higher (lower) values of $N^2$ right above it during the NH winter (summer) months, in agreement with the seasonal cycle of the tropopause (Yulaeva et al., 1994) and the tropical TIL climatological seasonal cycle described by Grise et al. (2010).

The daily evolution of $N^2$ also shows a secondary maximum below the zero wind line (bold black), at the easterly side of the descending westerly QBO phase, between 20-25km height. The zero wind line (of the descending westerly QBO phase) usually crosses the ∼20km level in summer, while crossing the ∼25km level in the earlier winter. This happens in 2002, 2004, 2006, 2008 and 2013, with the exception of 2010 when the zero wind line crosses ∼25km in summer. The enhanced $N^2$ is present under the zero wind line all the way from 35km altitude, but it is most evident in winter and spring (∼25km altitude and below). In winter and spring the secondary maximum of $N^2$ (red, about $8\times10^{-4}s^{-2}$) is close to the TIL strength (brown, about $9\times10^{-4}s^{-2}$ in the first kilometer above the tropopause), forming a double-TIL-like structure in static stability. However, this secondary maximum in $N^2$ shall not be viewed as a second TIL since it is quite far away from the tropopause. In the case of 2010, when the secondary maximum appears in summer it is much weaker, probably due to the fact that $N^2$ is generally weaker throughout the whole lower stratosphere in summer (Fig. 1). We also note that during the descending easterly QBO phase $N^2$ is enhanced above the zero wind line, again within the easterly wind regime. This time, the enhanced $N^2$ is much weaker than in the descending westerly QBO phase case, and only discernible in the lowermost stratosphere.

Grise et al. (2010) found a significant correlation of enhanced $N^2$ in the layer 1-3km above the TP when easterlies were present in the lowermost stratosphere, while no clear correlation was found in the 0-1km layer above the TP. From Fig. 1 we deduce that this correlation of the 1-3km layer originates in the secondary $N^2$ maximum found below the zero wind line of the descending westerly QBO (or above the zero line of the descending easterly QBO to a lesser degree, within the easterly QBO wind regime in any case). The QBO influence on the TIL, strictly the absolute $N^2_{max}$ that is found in the first kilometer above the TP, is hard to discern in Fig. 1. We investigate this in more detail in the subsection 3.3.

### 3.1.2 Horizontal structure of TIL strength

Figure 2 shows daily snapshots of TIL strength (sTIL, $N^2_{max}$) and collocated horizontal wind divergence (see section 2.3) between 30°S-30°N, together with 100hPa geopotential height contours, for four different days: two winter cases (left) and two summer cases (right), as examples representative of the variability in strength and zonal structures of the tropical TIL between 2007 and 2013.

The first remarkable aspect of the tropical TIL is that the magnitude of $N^2_{max}$ is much higher than in the extratropics. Values near the equator vary between 9-15 $\times10^{-4}s^{-2}$ and reach the $20\times10^{-4}s^{-2}$ mark sometimes, compared to a sTIL of $8$-$10\times10^{-4}s^{-2}$ generally found in polar summer or within

ridges in mid-latitude winter (Pilch Kedzierski et al., 2015). This can be attributed to the background temperature gradient in the lower stratosphere in the tropics, with a strong negative lapse-rate and a higher background lower-stratospheric $N^2$.

When a dipole of tropopause cooling and warming aloft (needed for TIL formation) is added to this background profile, the potential temperature gradient just above the tropopause increases

dramatically, giving the enormous $N^2_{max}$ values observed in Fig. 2.

The peak containing $N^2_{max}$ is very narrow and not always found at the exact same distance from the tropopause. Thus, when a zonal mean $N^2$ profile is computed, the high $N^2_{max}$ values get slightly smoothed out (Pilch Kedzierski et al., 2015). This is why the $N^2$ values in the first kilometer above the tropopause in Fig. 1 are lower than in Fig. 2.

As observed by Grise et al. (2010), in Fig. 2 we find that the strongest TIL is almost always centered at the equator, pointing towards equatorially trapped wave modes as TIL enhancers (which we analyze in section 4). When the sTIL zonal structure is compared to 100hPa divergence and geopotential height, it can be observed that higher $N^2_{max}$ is in general near regions of horizontally divergent flow (blue in Fig. 2) and a higher 100hPa surface. This high-low behavior with stronger-weaker TIL highly resembles the cyclone-anticyclone relationship with sTIL found in the extratropics (Randel

et al., 2007; Randel and Wu, 2010; Pilch Kedzierski et al., 2015).

So far, Figs. 1 and 2 confirm the vertical/horizontal structures of the TIL and its seasonality from earlier studies Grise et al. (2010); Kim and Son (2012), while reporting new features: the TIL relation with near-tropopause divergence (which we analyze next in section 3.2) and a secondary $N^2$ maximum above the TIL region driven by the QBO, whose influence on the TIL is analyzed in section

3.3.

### 3.2   Relationship with Divergence

In this subsection, we have a closer look at the relationship of the zonal structure of the tropical sTIL with its collocated horizontal wind divergence as shown in Fig. 2. For this, we bin sTIL and

tropopause height depending on the divergence value collocated with each GPS-RO observation, and make a mean within each divergence bin, as previous studies did with relative vorticity in the extratropics (Randel et al., 2007; Randel and Wu, 2010; Pilch Kedzierski et al., 2015). The resulting divergence versus sTIL diagrams are shown in Figure 3. In both summer and winter (Figs. 3 a, b) sTIL increases with divergence: from 11-12$\times 10^{-4}s^{-2}$ found with convergent flow (negative values)

or near-zero divergence, increasing steadily up to almost 15$\times 10^{-4}s^{-2}$ with increasingly divergent flow. The sTIL relation with divergence shown in the diagrams from Figs. 3 a and b is analogous to that of relative vorticity versus sTIL in the extratropics from earlier studies (Randel et al., 2007; Randel and Wu, 2010; Pilch Kedzierski et al., 2015). We also note that the $N^2_{max}$ in winter is always slightly higher than in summer for any divergence value, in agreement with the seasonality with

stronger TIL in winter from Fig.1 and the climatology by Grise et al. (2010). There is no clear link

between a higher tropopause and a stronger TIL. The variation of tropopause height with divergence is very small (see Appendix A, figure A1).

The relation of stronger TIL with divergent flow in Fig. 3 is consistent with the hydrostatic adjustment mechanism over deep convection described by Holloway and Neelin (2007), which results in a colder tropopause and an increased temperature gradient aloft. The hydrostatic adjustment mechanism is a dynamical response to compensate the pressure gradients created by a local tropospheric warming (latent heat release) from convection. The pressure gradients extend above the heating, and ascent and adiabatic cooling act to diminish these pressure gradients with height, cooling the tropopause region above the deep convective tower. Paulik and Birner (2012) showed that this negative temperature signal near the tropical tropopause can be found even a few thousand kilometers away from the convective region. In Fig. 3, we do not differentiate whether divergence is coupled to equatorial waves or not, so any type of convection would be included together for the TIL enhancement with divergent flow. The horizontal structures of sTIL in Fig. 2 can also be shaped by deep convection not related to equatorial waves.

Given the results with divergence from Figures 2 and 3, and the resemblance with the sTIL relationship with relative vorticity in the extratropics, the question arises whether the tropical and extratropical TIL could share the same enhancing mechanism. We postulate an affirmative answer.

The modelling experiments of Wirth (2003, 2004) showed that the stronger TIL in anticyclones in the extratropics was caused by two mechanisms: tropopause lifting and cooling (therefore the higher tropopause with anticyclonic conditions found by Randel et al. (2007); Randel and Wu (2010)); and vertical wind convergence above the anticyclone due to the onset of a secondary circulation between the cyclones and the anticyclones. In the tropics, the tropopause height effect is absent, but there is a clear relationship between sTIL and divergence (Fig. 3a and 3b). Such a horizontally divergent flow is coupled with vertical convergence for continuity reasons. Given that sTIL is rather constant with horizontally convergent flow, the TIL enhancement by vertical convergence in the tropics seems to come in hand with the aforementioned hydrostatic adjustment mechanism to deep convective outflow. We propose that vertical wind convergence near the tropopause is one mechanism enhancing the TIL at all latitudes, although caused by different processes: convection in the tropics and baroclinic waves in the extratropics.

Vertical wind convergence is related to anticyclones within baroclinic waves in the extratropics, but tropopause lifting (cooling) and the stratospheric residual circulation also enhance the TIL at the same time (Birner, 2010). In a similar way, we expect that the (100hPa) vertical wind convergence in the tropics is partly related to the equatorial wave spectrum, enhancing the tropical TIL along with other mechanisms (e.g. radiative forcing from water vapor or clouds). The equatorial wave modulation of the tropical TIL is studied in detail in section 4.

### 3.3 QBO influence

In Fig. 1 we showed that a secondary maximum of $N^2$ forms within the easterly wind regime of the QBO, just below the zero wind line of the descending westerly QBO phase, giving a double-TIL structure in the vertical $N^2$ profile in the lowermost stratosphere. This secondary $N^2$ maximum is responsible for the correlation of enhanced $N^2$ in the layer 1-3km above the tropopause with easterly winds (easterly QBO) found by Grise et al. (2010). However, no correlation was found in the layer 0-1km above TP (strictly where the TIL shall be), and no clear difference in TIL strength can be observed (Fig. 1) during the different phases of the QBO. We investigate this in more detail here, looking at the $N^2_{max}$ values found right above the TP instead of averaging over a certain layer.

To define the QBO phase, we take the zonal wind regime in the lowermost stratosphere, nearest to the TIL: around 18-20km altitude, which can be observed in Fig. 1 (black and dashed contour lines). We take two seasons of the same QBO phase in each case from the period between 2007-2013. The easterly phase of the QBO is found in the summers of 2007 and 2012 and the following winters of 2007/08 and 2012/13; while the westerly phase of the QBO is found in winters of 2008/09 and 2010/11 and the following summers of 2009 and 2011.

Figure 4 shows the distribution of $N^2_{max}$ for both winter (left, DJF) and summer (right, JJA). The black lines denote the average distribution over the 2007-2013 period, compared to easterly QBO phase (blue) and westerly QBO (red) as defined in the paragraph above. Winter has a higher mean $N^2_{max}$ ($12.17 \times 10^{-4} s^{-2}$) than summer ($11.39 \times 10^{-4} s^{-2}$), in agreement with the results of Grise et al. (2010) and Figures 1 and 3. We find that, during the easterly phase of the QBO (blue lines), the $N^2_{max}$ distributions slightly narrow and shift to lower values, giving lower seasonal means of $12.06 \times 10^{-4} s^{-2}$ in winter and $10.92 \times 10^{-4} s^{-2}$ in summer. During the westerly QBO phase (red lines), the opposite happens: the $N^2_{max}$ distributions widen and shift to higher values compared to the average distribution, giving higher seasonal means of $12.74 \times 10^{-4} s^{-2}$ in winter and $11.49 \times 10^{-4} s^{-2}$ in summer.

In both winter and summer, the seasonal mean $N^2_{max}$ in the westerly QBO phase is $\sim 0.6 \times 10^{-4} s^{-2}$ higher than during the easterly QBO phase. This difference is highly significant: the standard deviation of the seasonal mean is of the order of $0.02 \times 10^{-4} s^{-2}$ (each distribution's sample size is $\sim 20000$ profiles), and a t-test (two-tailed distributions with different sample sizes and variances) with these values gives us a t value of $\sim 20$, which is well beyond the 99.9 percent confidence level (critical value $\sim 3.3$).

In summary, from Fig. 4 we conclude that the tropical TIL is stronger during the westerly QBO phase in the lowermost stratosphere. This is not related to changes in the divergence distribution (given the relationship shown in Fig. 3), and is also anticorrelated with the strength of the secondary $N^2$ maximum found above the TIL region (Fig. 1).

The reason for this behavior of stronger TIL with westerly QBO is probably the modulation by Kelvin waves (the dipole of cooling near the tropopause and warming above), which have a higher

activity in the lowermost stratosphere with westerly shear, and a slower vertical propagation (Randel and Wu, 2005) which translates into a longer residence time and longer modulation near the tropopause region to enhance the TIL. How equatorial waves modulate the vertical temperature and $N^2$ structure is explained below in section 4.

## 4 Modulation by Equatorial Waves

### 4.1 Effect on the zonal structure of tropopause height

This section describes how equatorial waves modulate the temperature and $N^2$ vertical structure in the tropics. As explained in section 2.4, the gridded temperature and $N^2$ fields are filtered independently with 12 different bandpass filters in the wavenumber-frequency domain. As in Wheeler and Kiladis (1999), no bandpass filters overlap in the wavenumber-frequency domain (see Table 1), and the 12 filters amount for Kelvin waves, equatorial Rossby waves, MJO, and the three modes $n = (0,1,2)$ of westward-propagating (negative wavenumber $s$) Inertia-Gravity waves ($WIG_n$), the eastward-propagating ones (positive $s$, $EIG_n$), and the zonal wavenumber zero $_{s=0}IG_n$. For each wave type, a daily longitude-height section with its signature on temperature and $N^2$ is obtained.

Figure 5 shows examples of longitude-height snapshots with the $N^2$ anomalies ($\Delta N^2$) of Kelvin (Fig. 5a), Rossby (Fig. 5b), $EIG_n$ (Fig. 5c) and MJO band (Fig. 5d) at selected dates when the zonal structure of the tropopause (thick black line) is affected by the wave anomalies in an obvious way. In the case of $EIG_n$ the three modes $n = (0,1,2)$ are superimposed: the resulting field is $EIG_n = EIG_0 + EIG_1 + EIG_2$. Note that the filtered $N^2$ anomalies include a wide range of wavenumbers, 1-14 in the case of Kelvin waves for example (see Table 1), so the signatures of planetary waves 1 or 2 as well as transient shorter waves (higher wavenumbers) are represented together, giving a patchy appearance sometimes. Nevertheless, clear and coherent structures of the waves' $N^2$ signatures can be observed in Figure 5.

Temperature perturbations from Kelvin waves were observed to have their maximum near the tropopause in the study by Randel and Wu (2005). In Fig. 5 (with $N^2$), we see the same for all wave types: their maximum amplitude is generally found near and above the tropopause (black line). Also, zonal variations in tropopause height tend to be aligned with the wave's structure, with $N^2$ positive anomalies above the tropopause and negative anomalies below. This tropopause adjustment happens where the anomaly's amplitude is large, and is consistent with a dipole of tropopause cooling and warming aloft. This is clearly evident in Fig. 5a within 0-75°E and 100-180°E for Kelvin wave anomalies and in Fig. 5b within 0-125°W and ∼50°E for equatorial Rossby wave anomalies.

Although usually one wave type is dominant (therefore the choosing of separate dates in Fig. 5), different waves can influence the zonal structure of tropopause height at the same time: in Fig. 5 c and d (both of the same day, 2013-08-28), the MJO band creates a zonal variation of tropopause height within 50-125°E, while $EIG_n$ wave anomalies do so within 50-150°W. We note that in most

of the cases the strongest wave signatures, as well as zonal variations of the equatorial tropopause, are caused by transient waves with higher wavenumbers (as in every panel in Fig. 5) rather than planetary, quasi-stationary waves 1 or 2.

It is worth highlighting the structures that appear in the MJO band, with large amplitudes near the tropopause. Their eastward propagation, speed and longitudinal location match with the described patterns of deep convection associated to the MJO (Madden and Julian, 1994), but the horizontal and vertical scales are shorter. The $N^2$ anomalies from Fig. 5d are very similar to the composite temperature anomalies from the MJO band in the study by Kim and Son (2012), who found that MJO temperature anomalies near the tropopause have higher wavenumber due to their longer persistence compared to OLR anomalies.

When daily anomalies without running means are obtained as with our method, we see that the wave's anomalies shape the zonal structure of the tropopause, apart from the tropical tropopause layer (TTL) temperature and $N^2$ variability.

### 4.2 Average effect on the seasonal, zonal-mean profile

As explained in section 4.1, Figure 5 shows that the tropopause adjusts to the horizontal and vertical structure of the different wave types, with positive $N^2$ anomalies tending to be placed right above it. A tropopause-based mean of the wave's signature then should show an average enhancement of the TIL. The contribution of each equatorial wave type to the enhancement of the TIL is shown in Figure 6: for each wave type, a tropopause-based mean of the temperature and $N^2$ anomalies is done for all longitudes and winter days, achieving the average wave's effect on the seasonal, zonal-mean tropopause-based profile. $EIG_n$ (green), $WIG_n$ (orange) and $_{s=0}IG_n$ (grey) are the sum of all their modes $n = (0,1,2)$.

In Fig. 6a, all waves produce an average maximum cold anomaly right at the lapse-rate TP and a warm anomaly around 1-2km above the tropopause. Our results are in agreement with the study by Grise and Thompson (2013) that showed a cooling effect near the climatological tropopause by equatorial planetary waves, and remind of the dipole with tropopause cooling and lower-stratospheric warming found by Kim and Son (2012) which they attributed to convectively coupled waves. In the study by Kim and Son (2012), Kelvin waves and the MJO were the dominant wave types in short-term TTL temperature variability. By deriving daily fields with no temporal smoothing (see section 2.4.1), we were able to ascertain the role of waves with higher frequencies and zonal wavenumbers than previous studies, pointing out new important features from Fig. 6a: 1) the cold anomaly is maximized and centered right at the thermal tropopause, 2) all equatorial wave types give a similar signature, whose magnitude is dependent on the amount of the wave's activity, and 3) the role of transient waves with higher zonal wavenumbers and frequencies is significant: $WIG_n$, $EIG_n$ and Rossby waves have a bigger impact than the MJO.

The resulting $N^2$ signature (Fig. 6b) is a maximum $N^2$ enhancement right above the tropopause, and two regions of destabilization: below the tropopause and 2-3km above it. The overall effect is a TIL enhancement tightly close to the thermal tropopause. In Fig. 5 we showed obvious examples of tropopause adjustment to the wave structure with positive $N^2$ anomalies right above the tropopause. Given that the signature in the seasonal zonal-mean profile is considerable in Fig. 6, it can be concluded that the tropopause adjustment to the different waves (and the resulting dipole of colder tropopause / warm anomaly above in the tropopause-based zonal mean profile) occurs continuously, but not always so clearly as in Fig. 5. We stress that the signatures seen in Figures 5 and 6 were obtained by filtering the temperature and $N^2$ fields directly and independently, without any filtering in the vertical dimension.

Looking at the different wave types separately in Fig.6, the Kelvin wave (blue) has the strongest temperature signature (Fig. 6a), but owing to its longer vertical scale (see Fig. 5a) the temperature gradient that the Kelvin wave produces is closer to the rest of the waves', giving an average $N^2$ enhancement of $0.35 \times 10^{-4} s^{-2}$. The signatures of $EIG_n$ (green), $WIG_n$ (orange) and Rossby (red) waves give an average TIL enhancement of $\sim 0.25 \times 10^{-4} s^{-2}$ each, followed by the MJO band (purple, $0.1 \times 10^{-4} s^{-2}$). The $_{s=0}IG_n$ wave type (grey), although lacking zonal structures by definition and having little activity, still gives a minor TIL enhancement. The relative $N^2$ minima below the tropopause and above the TIL region in the seasonal zonal-mean profile (e.g. Grise et al. (2010)) can be attributed to the equatorial wave modulation as well.

The total effect of the equatorial waves on the equatorial zonal-mean seasonal temperature profile is a $\sim 1.1$K colder tropopause and a $\sim 0.5$K warm anomaly above, with a resulting TIL enhancement of $\sim 1.2 \times 10^{-4} s^{-2}$. Fig. 6 shows the mean wave effect during winter; the results are similar during summer (see Appendix B, figure B1) except for a slightly weaker effect of the MJO band, given its lower average activity in that season.

We acknowledge the possibility that the wave signals shown in Figs. 5 and 6 may not be 100% dynamical: a radiative component is included if clouds are present near the tropopause (radiative cooling at the cloud top that creates a temperature inversion). Part of the equatorial wave spectrum in the TTL is known to be coupled with convection, a small part of which reaches the tropopause (Wheeler and Kiladis, 1999; Fueglistaler et al., 2009), and the occurrence of cirrus clouds is also related to equatorial waves (Virts and Wallace, 2010). Case studies using GPS-RO data have investigated the temperature inversion generally found at cloud tops, for convective clouds (Biondi et al., 2012) and non-convective cirrus clouds (Taylor et al., 2011). The signal from any cloud coupled with an equatorial wave would be captured by its corresponding wavenumber-frequency domain filter, since the cloud signal would travel together with the wave in the same domain. Therefore a part of the mean wave signal shown in Fig. 6 could be due to the temperature inversion of (wave-coupled) cloud-tops near the tropopause, but quantifying this is beyond the scope of our study. Nevertheless, it is logical to assume that the radiative part associated to the equatorial wave signal shall be small,

since near-tropopause height cloud tops are not frequent, equatorial waves are not radiatively driven
       and their propagation is explained by dry dynamics.

       Also note that in the case where the equatorial waves (Figs. 5 and 6) are coupled to convection,
       the tropopause cooling by the hydrostatic adjustment mechanism (Holloway and Neelin, 2007) is
       captured by the filters as well, and a much refined methodology would be needed to separate the
contribution of equatorial waves and convection alone.

       Our results from this section (Figs. 5 and 6) agree with earlier studies that derived equatorial
       wave anomalies from GPS-RO data (the Kelvin and MJO signatures and their amplification near the
       tropopause (Randel and Wu, 2005; Kim and Son, 2012)). Also, Fig. 6 confirms the effect of equato-
       rial waves on the mean temperature profile (colder tropopause and warm anomalies above forming a
dipole (Kim and Son, 2012; Grise and Thompson, 2013)) and their crucial role in enhancing the TIL
       in the tropics (Grise et al., 2010). The novelty in our study resides in that we include small-scale,
       higher-frequency waves (e.g. Inertia-Gravity waves); and that we are able to quantify the effect of
       each equatorial wave type separately by tropopause-based averaging of the filtered wave anomalies.

       Our results from this section should not be viewed as a mere quantification of the waves them-
selves or an artifact of the tropopause-coordinate. Although transient and instantaneous, there are
       motions associated to the wave signals that locally lift/cool/modulate the tropopause, and also warm
       the air aloft. Another characteristic of the waves is that they amplify next to and above the tropopause
       (Fig. 5), and also increase their vertical tilt (Fig. 5a, visible for Kelvin waves), which increases
       the wave signal in the TIL region, and also increases the area of positive $N^2$ anomaly above the
tropopause. This is a response of the wave to the elevated $N^2$ values in the lowermost stratosphere,
       in agreement with linear theory, which in turn enhances the TIL further, working as a positive feed-
       back. Although our results point in this direction, more research needs to be carried out to consider
       such a feedback as a robust feature of the global tropopause region.

## 4.3   TIL without equatorial wave signals

Figure 7 shows the daily evolution of the equatorial zonal-mean, tropopause-based $N^2$ profile (Fig.
       7a), and the resulting $N^2$ profile when the equatorial wave signals are subtracted (Fig. 7b). The
       display is very similar to that of Fig. 1, but in order to allow the subtraction of the equatorial wave
       signal, for Fig. 7 we use the gridded dataset obtained in section 2.4, from COSMIC profiles only
       (2007-2013) and 10°S-10°N without any temporal smoothing of $N^2$.
A clear difference in the TIL region can be observed in Fig. 7b: without the equatorial wave
       signal, the TIL in the first kilometer above the tropopause is much weakened, from $N^2$ values of
       7-9×10$^{-4}s^{-2}$ right above the tropopause (orange-red colors in Fig. 7a) to values of 6-7×10$^{-4}s^{-2}$
       (yellow-orange colors in Fig. 7b) and even less sometimes. In Fig. 7b, the stronger TIL with $N^2$
       values above 7×10$^{-4}s^{-2}$ (red) is very sparse in time and restricted to wintertime. The differences

between Fig. 7a and Fig. 7b agree well with the magnitude of mean TIL enhancement calculated in section 4.2 (Fig. 6).

However, in Fig. 7b the deeper $N^2$ structures between the tropopause and $\sim$20km altitude remain intact, as well as the secondary $N^2$ maximum below the descending westerly QBO phase: they are basically the same as in Fig. 7a and therefore not directly modulated by equatorial waves.

What other mechanisms could enhance the TIL in the tropics? Deep convection that is not coupled with any equatorial wave can also lead to tropopause cooling (by hydrostatic adjustment) and TIL enhancement, as discussed in section 3.2 (Holloway and Neelin, 2007; Paulik and Birner, 2012). Given that deep convection near the equator is more frequent in winter, this would explain the occurrence of stronger TIL in Fig. 7b within this season. Radiative cooling from non-convective cloud
tops near the tropopause (e.g. Taylor et al. (2011)), or from strong humidity gradients across the tropopause, can also enhance the gradients that lead to TIL enhancement.

Also note that the wave signals in Fig. 5, their average signature in Fig. 6, and the subtracted signals in Fig. 7b, all come from the instantaneous filtered anomalies: once the wave has left the tropopause region, or dissipated, our filters do not capture any signal that could modulate the TIL.
The wave-mean flow interaction is not visible with our method, since its more persistent temperature and $N^2$ effect would not travel in the wavenumber-frequency domain any more.

The secondary $N^2$ maximum below the descending westerly QBO phase can be related to the temperature anomaly associated to the wind shear of the QBO (Baldwin et al., 2001), which affects the background $N^2$ structure throughout the stratosphere. It can be seen in Figs. 1 and 7 that dur-
ing the easterly phase of the QBO, $N^2$ between 20-30km altitude is generally higher than within westerlies. It is also possible that the $N^2$ maximum right below the zero wind line of the descending westerly QBO could be forced by a temperature anomaly from the dissipation of Kelvin waves, which propagate vertically with easterlies until they reach westerly shear. In this case, it would be an indirect effect of Kelvin waves: once they dissipate there is no signal to be captured by our fil-
ter. Quantifying this effect (for both the secondary $N^2$ maximum and the TIL, as in the previous paragraph) is beyond the scope of our study.

## 5 Discussion: Applicability of the Wave Modulation in the Extratropics

As Fig. 6 showed, all equatorial wave types have the same effect on the temperature and $N^2$ seasonal zonal-mean vertical profiles, only varying in magnitude. Since all wave types have the same
signature, one could expect a similar picture coming from the extratropical wave spectrum. Taking the extratropical baroclinic Rossby wave as an example: the embedded cyclones-anticyclones with lower-higher tropopause would be an example of tropopause adjustment to the anomalies associated with the wave, as in Fig. 5. Given that the zonal variability of $TP_z$ at mid-latitudes is much larger than within the tropics (3km against 0.8km, see Appendix A) and that temperature gradients next to

the jet stream are also of much higher magnitude, it is probable that the extratropical Rossby wave's $N^2$ signature on the mid-latitude zonal mean profile is even stronger than the signal observed from Kelvin waves in Fig. 6, which dominates in the tropics. Inertia-gravity waves are also widely present in the extratropics. Depending on the amplitude they reach next to the extratropical tropopause, this wave type shall also contribute to enhance the TIL, which is predicted by the modelling experiment

by Kunkel et al. (2014).

The wavenumber-frequency domain filtering method used with the dispersion curves of extratropical wave modes would be suited to quantify the modulation of each wave mode on the extratropical TIL, in the same way our study has done with equatorial waves. Also, similarly to section 4.3 and Fig. 7, it could be possible to show how much of the TIL in the extratropics is due to processes

other than the instantaneous modulation by extratropical waves (i.e. radiative forcing or residual circulation). Preliminary results show that the method used in this paper is indeed applicable in the extratropics as well, and a new paper about this is in preparation.

## 6  Concluding Remarks

Our study explores the horizontal and vertical variability of the tropical TIL, the effect of the QBO,

the role of near-tropopause horizontal wind divergence and the role of equatorial waves in enhancing the tropical TIL. Overall it gives an in-depth observational description of the TIL properties in the tropics and the mechanisms that lead to its enhancement in a region where research has focused very little so far.

Our results agree with the seasonality and location of the tropical TIL described by Grise et al.

(2010), with stronger TIL centered at the equator and peaking during NH winter. We describe a new feature: a secondary $N^2$ maximum that forms above the TIL region within the easterly wind regime of the QBO, below the zero wind line of the descending westerly QBO (Fig. 1). This secondary maximum leads to a double-TIL-like structure in the stability profile, and explains the correlation of enhanced $N^2$ in the 1-3km layer above the tropopause with easterly QBO found by Grise et al.

(2010). The behavior of the secondary $N^2$ maximum is anticorrelated with the TIL strength (strictly the $N^2_{max}$ found less than 1km above the tropopause): the TIL is stronger during the westerly phase of the QBO in the lowermost stratosphere (Fig. 4).

The zonal structure of the tropical TIL shows a stronger (weaker) TIL with near-tropopause divergent (convergent) flow (Fig. 2). This sTIL-divergence relationship (Fig. 3) is analogous to that of TIL

strength with relative vorticity found in the extratropics (Randel et al., 2007; Randel and Wu, 2010; Pilch Kedzierski et al., 2015), and we suggest that vertical wind convergence is a TIL enhancing mechanism that the tropics (divergent flow) and extratropics (anticyclones) have in common.

We also quantified the signature of the different equatorial waves on the seasonal zonal-mean temperature and $N^2$ profile (Fig. 6). All wave types have, on average, maximum cold anomalies at

the thermal tropopause and warm anomalies above, enhancing the TIL strength very close to the tropopause. The way this modulation is done is by tropopause adjustment to the vertical structure of the wave's associated anomalies when these have high amplitudes (Fig. 5). While agreeing with earlier studies that used GPS-RO to investigate equatorial waves (Randel and Wu, 2005; Kim and Son, 2012), our results show the importance of small-scale, high-frequency waves due to our method with

minimized temporal smoothing, which enables us to quantify and compare the role of each different equatorial wave type for the first time. Inertia-gravity and Rossby waves play a very significant role, with a bigger signature than the MJO, and Kelvin waves dominate the net tropopause cooling and warming above in the tropopause-based profile, with the resulting TIL enhancement (Fig. 6).

Without the equatorial wave signal, the TIL is much weakened (Fig. 7) but part of it remains,

and we point to non-wave-coupled deep convection (tropopause cooling by hydrostatic adjustment, (Holloway and Neelin, 2007; Paulik and Birner, 2012)) and radiative effects from clouds or humidity gradients as other possible mechanisms that could enhance the tropical TIL.

We suggest that this wave modulation will also be present in the extratropics with baroclinic Rossby and inertia-gravity waves as main contributors, which will be the subject of a follow-on

study.

**Appendix A: Divergence vs Tropopause Height**

Figure A1 shows divergence versus tropopause height ($TP_z$) diagrams, as in Fig. 3 with TIL strength (section 3.2). There is no clear link between a higher tropopause and a stronger TIL. In summer (Fig. A1 a), the relation of higher $TP_z$ with divergent flow is very small: the difference between

convergent-divergent flow is only of 0.8km, while the difference in cyclones-anticyclones in the extratropics is over 3km (Randel and Wu, 2010). In winter (Fig. A1 b) this relation is non-existent: the tropical $TP_z$ is around 17km at all divergence-convergence values.

**Appendix B: Equatorial Wave Modulation in Summer**

Figure B1 is the summer counterpart of Fig. 6 from section 4. The average effect of each wave in

summer (Fig. B1) is very similar to the one from winter (Fig. 6), except for a smaller MJO signature.

**Appendix C: Caveat on the Filtering of Periods of 1-2 Days from a Daily Dataset**

As shown in Table 1, the modes $n = 2$ of all inertia-gravity wave types ($EIG_n$, $WIG_n$ and $_{s=0}IG_n$) are defined for periods between 1-2 days. With a dataset of daily temporal resolution, filtering with such periods has to be taken with lots of caution for two reasons:

1) Oscillations with periods below 2 times the temporal resolution of the dataset (below 2 days in this case) are underestimated (best case scenario), or not resolved at all by the dataset. Nevertheless we applied these filters, should any part of the wave signal be discernible.

2) Once filtered, the resulting wave anomalies are subject to include spurious signals because of spectral ringing. It is very important to know how much of the wave signature in Figure 6 comes

from this artifact, since all modes $n = (0,1,2)$ are summed-up there.

We computed the mean signature of modes $n = (0,1,2)$ separately, and found that all of the signal in Fig. 6 comes from modes $n = 0$ and 1. This means that the filters of inertia-gravity waves with periods between 1-2 days do not capture any signal at all (artificial or not), and therefore make no contribution to our results. The equatorial wave signature in Fig. 6 (and Fig. B1) comes entirely from

oscillations that are resolved by our gridded daily dataset obtained from COSMIC GPS-RO profiles.

Waves with periods below 2 days could modulate the tropical tropopause region and the TIL, though: only the current amount of GPS-RO profiles is not enough to resolve this. It shall be possible to do so once COSMIC-2 profiles (a much increased amount compared to current data) are available.

*Acknowledgements.* This study was completed within the Helmholtz-University Young Investigators Group

NATHAN project, funded by the Helmholtz Association through the president's Initiative and Networking Fund and the GEOMAR Helmholtz-Centre for Ocean Research in Kiel. We thank the ECMWF data server for the freely available ERA-Interim data; and UCAR for the COSMIC, CHAMP and GRACE satellite missions' temperature profiles. Comments by Joowan Kim and two anonymous reviewers are highly appreciated and clearly helped to improve the manuscript significantly. The assistance accessing different datasets and discussions with

Sandro Lubis, Wuke Wang and Sebastian Wahl are also appreciated.

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

**Table 1.** Parameters used to bound the filter of the different equatorial wave types, with the meridional mode $n$ as subscript: $t$ (period, in days), $s$ (zonal planetary wavenumber) and $h$ (equivalent depth, in m)

| **Wave type** | $t_{min}$ | $t_{max}$ | $s_{min}$ | $s_{max}$ | $h_{min}$ | $h_{max}$ |
|---|---|---|---|---|---|---|
| Eq. Rossby | 6 | 70 | -14 | -1 | 6 | 600 |
| Kelvin | 4 | 30 | 1 | 14 | 6 | 600 |
| MJO | 30 | 96 | 2 | 5 | 8 | 90 |
| $WIG_0$ | 2.5 | 6 | -10 | -1 | 6 | 360 |
| $WIG_1$ | 2 | 2.5 | -15 | -1 | 8 | 90 |
| $WIG_2$ | 1 | 2 | -15 | -1 | 8 | 90 |
| $EIG_0$ | 2.5 | 4 | 1 | 15 | 8 | 90 |
| $EIG_1$ | 2 | 2.5 | 1 | 15 | 8 | 90 |
| $EIG_2$ | 1 | 2 | 1 | 15 | 8 | 90 |
| $_{s=0}IG_0$ | 3 | 6 | -0.1 | 0.1 | 8 | 90 |
| $_{s=0}IG_1$ | 2 | 3 | -0.1 | 0.1 | 8 | 90 |
| $_{s=0}IG_2$ | 1 | 2 | -0.1 | 0.1 | 8 | 90 |

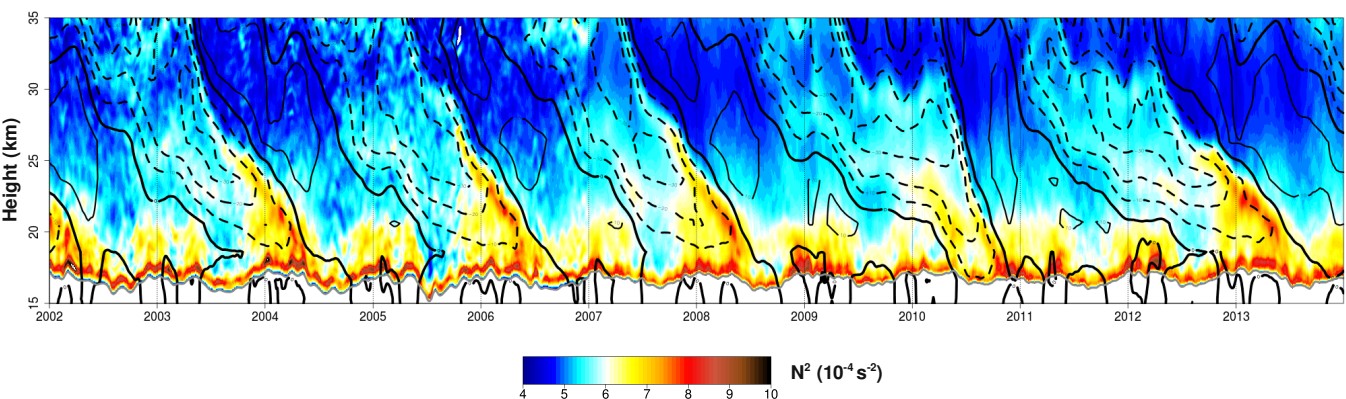

**Figure 1.** Daily evolution of the tropopause-based, equatorial (5°S-5°N) zonal mean $N^2$ vertical profile between 2002-2013 (colors). 2002-2006 from CHAMP+GRACE GPS-RO profiles, 2007-2013 from COSMIC. The grey line denotes the tropopause height ($TP_z$). Thin black contours denote positive (westerly) mean zonal wind, with a thicker contour for the zero line, dashed contours for negative (easterly) winds, and a 10m/s separation. To improve visibility, each day shows the running mean $N^2$ profile and $TP_z$ of +-7 days. In the case of the winds, the running mean is made for +-15 days.

**Winter cases**

**Summer cases**

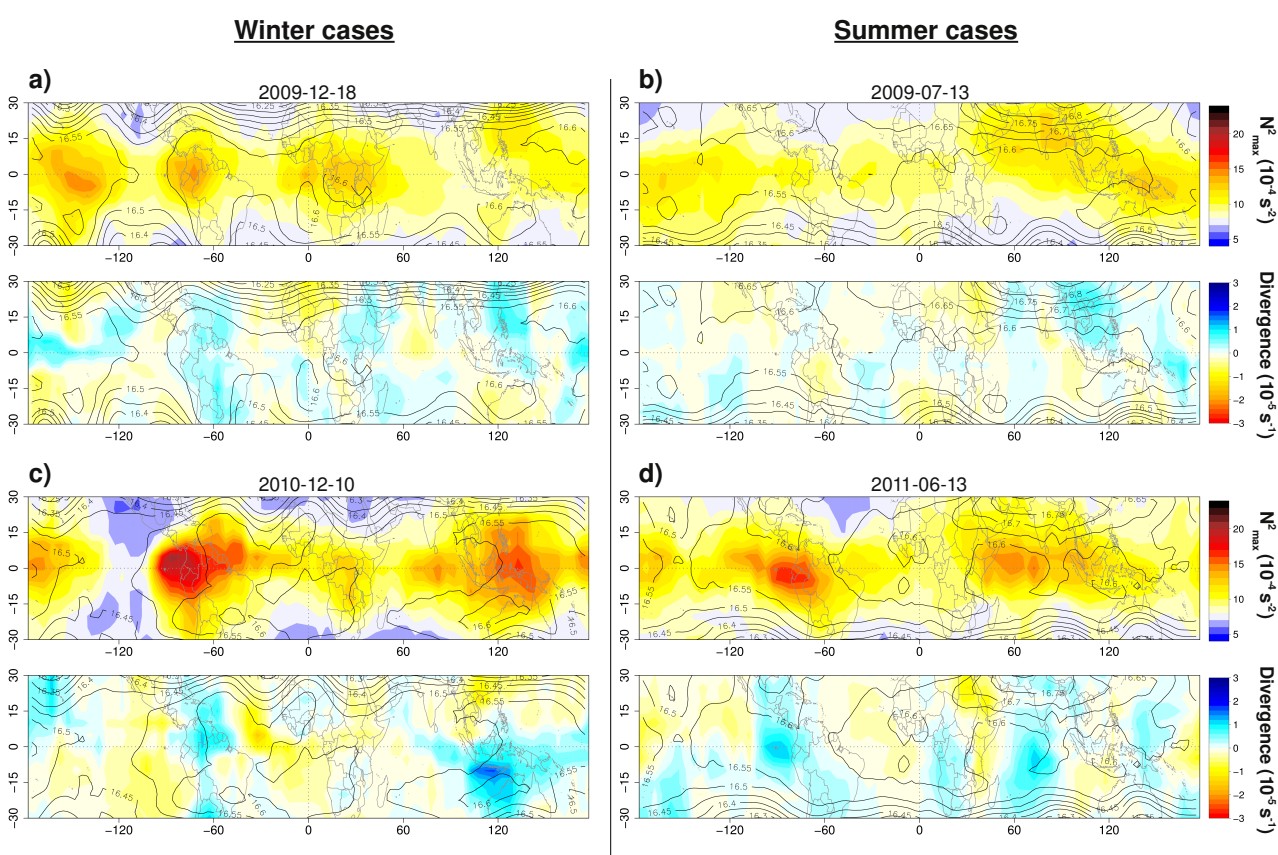

**Figure 2.** Maps of daily TIL strength ($N^2_{max}$, first and third rows) and 100hPa horizontal wind divergence (second and fourth rows). Winter cases are on the left side, and summer cases are on the right side. Corresponding color scales are on the right end. Contour lines show the 100hPa geopotential height (in km' with 50m' interval).

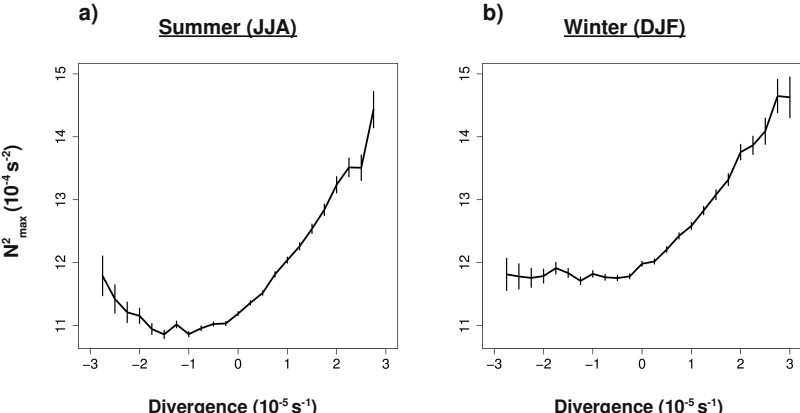

**Figure 3.** Diagrams of horizontal wind divergence versus TIL strength ($N_{max}^2$) for the latitude band 10°S-10°N. a) belongs to the summer season (JJA), and b) to the winter season (DJF). Vertical bars denote one standard deviation of the mean value.

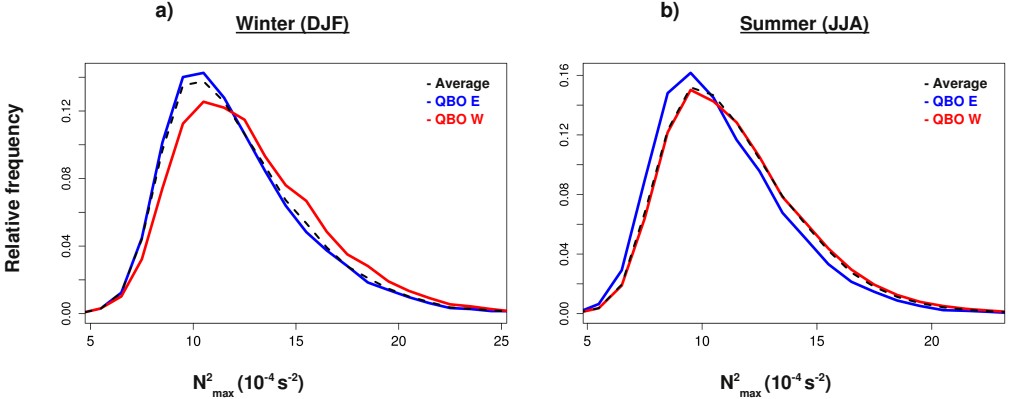

**Figure 4.** Histograms with relative frequency of TIL strength ($N_{max}^2$), for winter (a, DJF) and summer (b, JJA). The black dashed line denotes the average seasonal distribution. The blue line shows distributions during the easterly phase of the QBO in the lowermost stratosphere, and the red line shows the distributions during the westerly QBO phase.

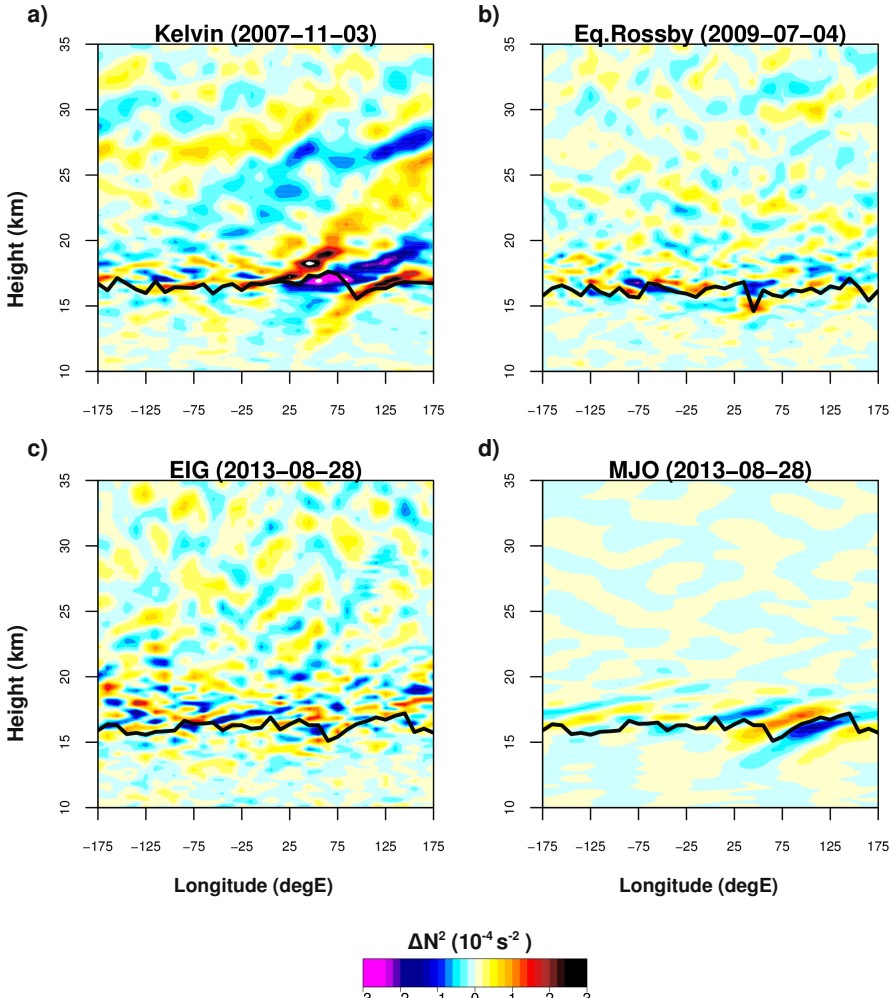

**Figure 5.** Longitude-height snapshots of static stability anomalies ($\Delta N^2$) of different wave types at certain dates: a) Kelvin wave, b) Rossby wave, c) Eastward IGW ($EIG_n$), d) MJO band. The black line denotes the thermal tropopause.

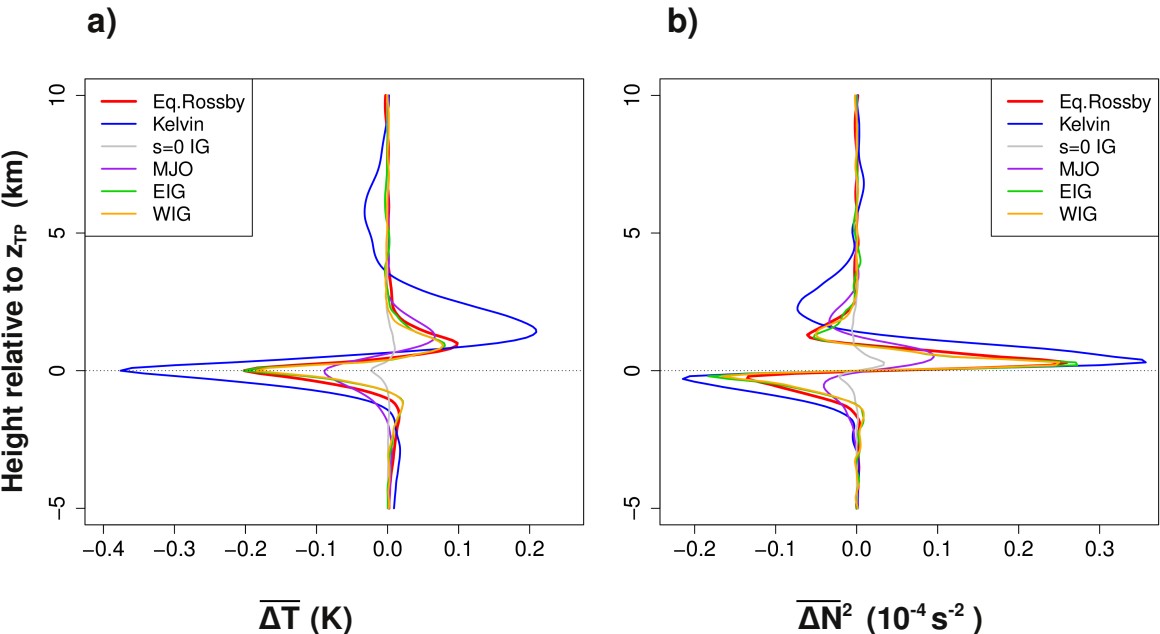

**Figure 6.** Winter (DJF) average signature of the different wave types, as the mean anomalies of (a) temperature $\overline{\Delta T}$ and (b) static stability $\overline{\Delta N^2}$ in the equatorial zonal-mean vertical profiles (10°S-10°N).

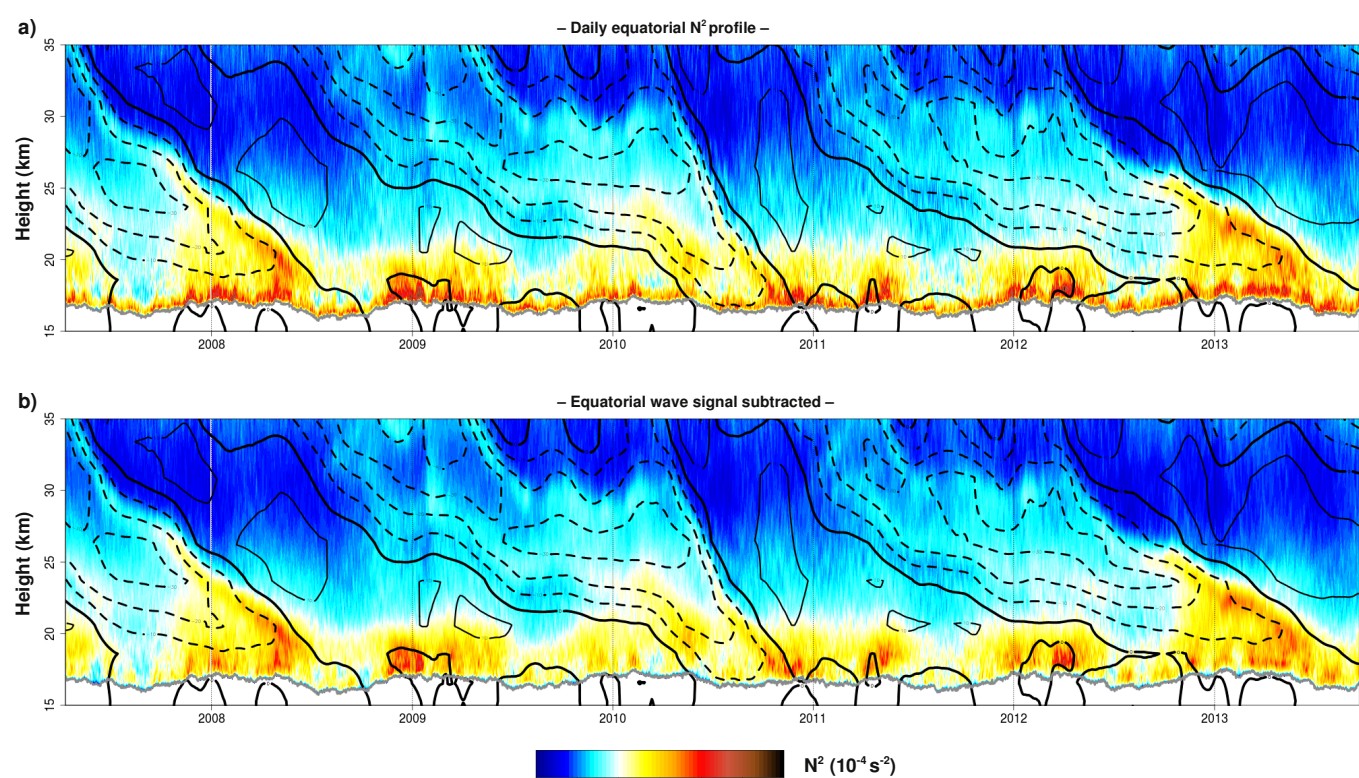

**Figure 7.** a) Daily evolution of the tropopause-based, equatorial (10°S-10°N) zonal mean $N^2$ vertical profile between 2007-2013 (colors) from COSMIC GPS-RO profiles. The grey line denotes the tropopause height ($TP_z$). Thin black contours denote positive (westerly) mean zonal wind, with a thicker contour for the zero line, dashed contours for negative (easterly) winds, and a 10m/s separation. To improve visibility, the winds are displayed with a running mean of +-15 days. No running mean is applied to the $N^2$ vertical profile or $TP_z$ in order to allow the subtraction of the equatorial wave signal. b) Equatorial wave signal subtracted from the $N^2$ vertical profile.

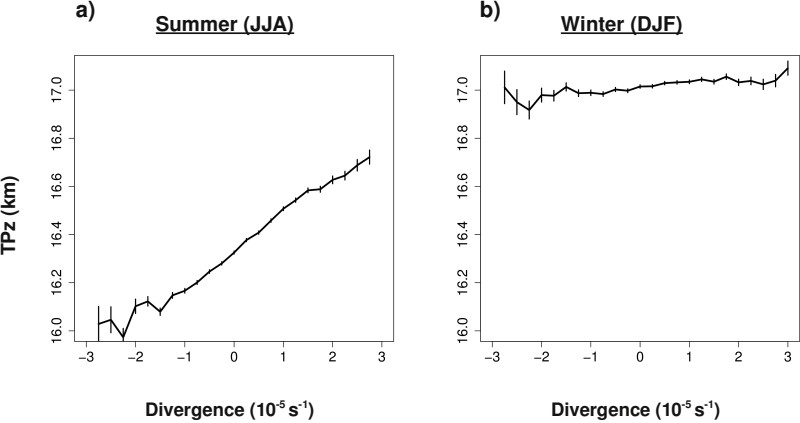

**Figure A1.** Diagrams of divergence versus tropopause height ($TP_z$, km) for the latitude band 10°S-10°N. a) belongs to the summer season (JJA), and b) to the winter season (DJF). Vertical bars denote one standard deviation of the mean value.

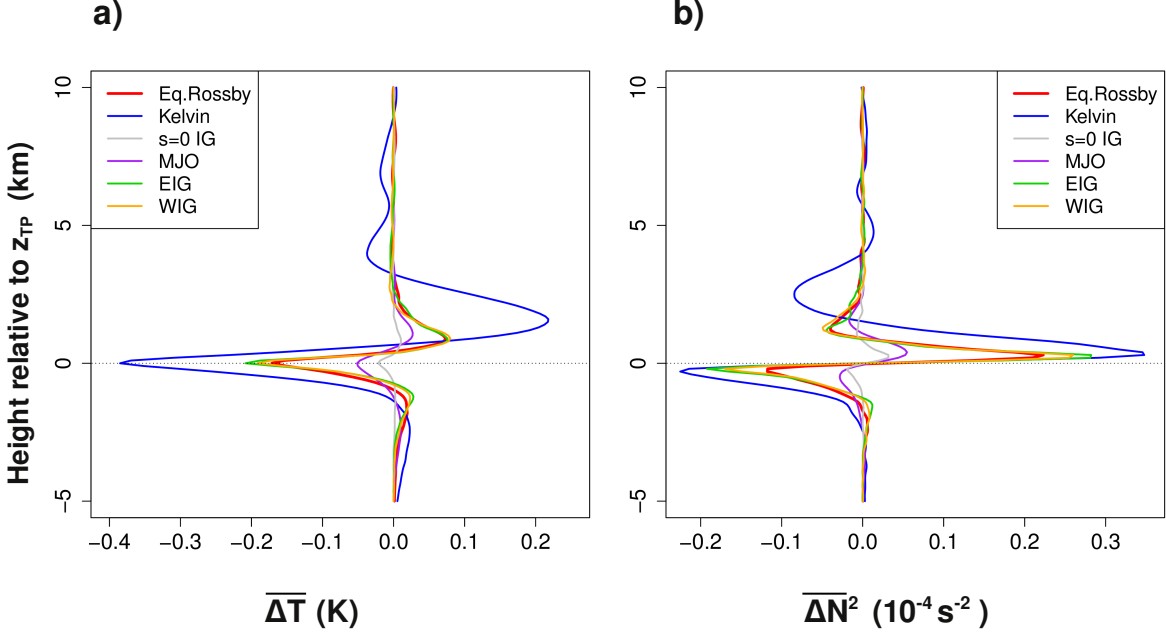

**Figure B1.** Summer (JJA) average signature of the different wave types, as the mean anomalies of (a) temperature $\overline{\Delta T}$ and (b) static stability $\overline{\Delta N^2}$ in the equatorial zonal-mean vertical profiles (10°S-10°N).