# Peer review of "The Tropical Tropopause Inversion Layer: Variability and Modulation by Equatorial Waves"

_Atmospheric Chemistry and Physics, 2016_

## Referee Comment (RC1) · Anonymous Referee #1 · 11 Apr 2016

General comment

The paper investigates the tropical tropopause inversion layer (TIL). Although this has been the subject of previous studies, the present work provides new knowledge by using a more detailed data set and by doing a more detailed analysis. The authors clearly state in their text what is new and what has been known before, so I think the work is put into context very well. Overall the paper is concise and well written. Nevertheless I have a few comment. These should help to produce a revised version, which I am sure would be a valuable contribution to ACP.

[Figure]

**Major issues**

1. I am not an expert on reanalysis data, but as far as I know the quality of reanalysis wind data in the tropics is not as good as one would wish them to be. So the question is: how much can you trust the upper tropospheric horizontal divergence in the tropics? The authors should at least address this issue and try to convince the reader that the quality of the data is sufficient for their purpose.

2. The authors could clarify the role of tropospheric vertical motion and upper tropospheric horizontal divergence for tropical TIL formation, e.g. in their section 3.2. Assuming that a tropospheric wave produces regional upwelling with horizontal divergence right at the tropopause level, this would yield a higher and sharper than normal tropopause (corresponding to a stronger than normal TIL) — essentially by pushing upward the tropopause and thereby making the lowermost stratosphere somewhat colder. In this simple scenario there is *no warming* involved at any point: the TIL forms because the *cooling* ($\partial T/\partial t < 0$) has some vertical structure decaying with altitude. On the other hand, composite plots like Figure 6a indicate actually some *warming* in the lowermost stratosphere. Does this mean that the equatorial waves are associated with *downwelling* in the lowermost stratosphere (right above the tropospheric upwelling), or does this possibly imply diabatic warming?

**Minor issues**

1. Line 166: What is an e-fold function? A Gaussian?

2. Line 172: How are the profiles shifted in altitude? By how much? For what purpose?

3. If I recall right, an important point in the work of Wheeler and Kiladis (1999) is the removal of the background spectrum. How is this dealt with in the present work?

4. As a standard reference for the seasonal cycle of the tropical tropopause one should add the paper by Yulaeva *et al.* (1994).

5. Line 258, ".... temperature inversion is added to this background profile...": For me, "temperature inversion" means that the temperature increases (rather than decreases) with altitude. It seems that this term should only be used for full temperature profiles, not for perturbations or "additions". So I have a difficulty with the expression "adding a temperature inversion to the background profile".

6. Line 259: "skyrocket" appears too colloquial and not quite fitting here.

7. Line 260, "the $N^2_{max}$ is very narrow": strictly speaking this is not true. The peak containing $N^2_{max}$ may be very narrow, not the $N^2_{max}$ itself.

8. Line 336: How is the significance of the difference between the curves assessed? As far as I know, the significance of the difference in the mean between two distributions is measured by the *standard error* (Press *et al.*, 1992), not by the standard deviation.

9. Line 364: How was the longitude chosen for the plots in figure 5?

10. Line 374, "... tend to be aligned...": Well, this seems to be at least partly wishful thinking, I find that it is sometimes true, but sometimes not.

11. Line 378, "... cooling and/or warming...": this is not clear to me.

12. Line 396 and line 401: Figure 5 shows anomalies of $\partial N^2/\partial t$, not anomalies of $T$!

13. Line 457, "... a small part...": how do you know that this part is small? Could it be a substantial part?

14. Line 485, should read: "... would be suited to....".

15. Line 525, "... is rather marginal...": "marginal" may not be the right term here. True, it is smaller than in the corresponding figure 3, but it may yet be significant!

**References**

Press, W. H., B. P. Flannery, S. A. Teukolsky, and W. T. Vetterling 1992. *Numerical Recipes. The Art of Scientific Computing*. Cambridge University Press, 2nd edition, 818 pp.

Yulaeva, E., J. R. Holton, and J. M. Wallace 1994. On the cause of the annual cycle in tropical lower-stratospheric temperatures. *J. Atmos. Sci.* **51**, 169–174.

---

## Referee Comment (RC2) · J. Kim (Referee) · 17 Apr 2016

General comments

This paper examines the tropopause inversion layer (TIL) in the tropics using GPS radio occultation temperature measurements. The coherent behaviors of static stability (N2; also TIL strength) with the Quasi-Biennial Oscillation (QBO) and near-tropopause divergence are clearly demonstrated. The equatorial waves and their signatures in temperature and N2 are also investigated in order to explain the role of the equatorial waves in setting TIL strength. This manuscript is generally well organized and successfully demonstrates the fine-scale feature of the TIL. Although there are couple of minor issues that should be addressed before publication, this paper is recommended for publication in ACP.

Some minor issues are listed below.

Specific comments (minor)

1. The title is too broad for the contents of the manuscript. Authors are mainly focusing on dynamical mechanisms that could enhance TIL in the tropics. Although they demonstrate the mechanisms clearly, the contents in the manuscript are still too limited to cover the whole spectrum of the tropical TIL (e.g., annual cycle, influence of deep convection and radiation, role of shallow Brewer-Dobson circulation). It is strongly recommend for authors to further specify the title of this manuscript.

2. Authors suggest that Kelvin waves cause the enhancement of N2 just below the westerly shear (or zero-wind line) of the QBO. This may be one possible cause, however, the zonal mean temperature anomaly associated with vertical wind shear of the QBO (cf. Fig. 4 in Baldwin et al. 2001) has a strong impact on N2. Several Kelvin of temperature changes in ∼10 km depth, and this could significantly modulate N2 in the lower stratosphere. In fact, this may have a bigger impact on N2 than Kelvin waves, particularly in zonal mean field. Some analysis and discussion on this effect will be helpful (a simple comparison of tropical mean temperature profiles in westerly and easterly QBO will be good enough).

3. Although influence of deep convection on TIL is beyond the scope of this study, some discussions on tropical convection will still be helpful. For example, the zonal structures in N2 (shown in Fig. 2) are largely related to deep convection in DJF and JJA. In fact, climatology of N2 shows similar structures as in Fig. 2, and this is largely due to tropopause cooling cause by deep convection (deep convection make tropopause colder; e.g., Johnson and Kriete 1982; Gettelman et al. 2002; Paulik and Birner 2012). Only a part of the N2 structure is explained by tropical waves.

In addition, the coherence between N2 and near-tropopause divergence (which is a noble contribution of this paper) is consistent with the hydrostatic adjustment mechanism, which is proposed by Holloway and Neelin (2007) to explain cold-top (tropopause) over deep convection. Those discussion could be helpful for readers.

Technical suggestions

Line 35: Satellite GPS => Global Positioning System (GPS)

(In many place, satellite GPS => GPS)

Line 105: tropopause height (TPz) using the WMO lapse-rate tropopause criterion...

Line 169: latitude (y) and time (t). The maximum distance allowed from the grid point in each dimension is 5°longitude, 10°longitude, and 12 hours, respectively.

Line 214: 3.1 ?

Line 234: 2011=>2010?

Line 375: highest amplitude => maximum amplitude

Line 379: very high => very large

Line 393: high amplitude => large amplitude

Line 476: higher that within => larger than that in

Fig 5: why do you show N2 tendency (dN2/dt) instead of N2?

(also in Fig 6: dT/dt instead of T)?

References

Baldwin, M. P., and Coauthors, 2001: The Quasi-Biennial Oscillation. Rev. Geophys., 39, 179–229.

Johnson, R. H., and D. C. Kriete, 1982: Thermodynamic and Circulation Characteristics of Winter Monsoon Tropical Mesoscale Convection. Mon. Wea. Rev., 110, 1898–1911, doi:http://dx.doi.org/10.1175/1520-0493(1982)110<1898:TACCOW>2.0.CO;2.

Gettelman, A., M. L. Salby, and F. Sassi, 2002: Distribution and influence of convection in the tropical tropopause region. J. Geophys. Res., 107, doi:10.1029/2001JD001048.

[Figure]

Paulik, L. C., and T. Birner, 2012: Quantifying the deep convective temperature signal within the tropical tropopause layer (TTL). Atmos. Chem. Phys., 12, 12183–12195, doi:10.5194/acp-12-12183-2012.

Holloway, C. E., and J. D. Neelin, 2007: The Convective Cold Top and Quasi Equilibrium. J. Atmos. Sci., 64, 1467–1487, doi:10.1175/JAS3907.1.
* * *

---

## Referee Comment (RC3) · Anonymous Referee #3 · 10 May 2016

The tropopause inversion layer (TIL) in the tropics is studied using temperatures from GPS radio occultations and dynamical fields from ERA-interim. A strong relationship of TIL strength to tropopause-level divergence is found, with stronger divergence associated with stronger TIL. The authors also analyze the modulation of the TIL strength by the QBO and equatorial waves. They find a stratification maximum located just below the transition line from easterly to westerly QBO phase, which becomes a secondary TIL when this transition approaches the tropopause. Equatorial waves are found to strongly modulate temperatures around the local tropopause (as in previous work in the literature), thereby leading to transient local TIL modulation.

As the authors remark, the tropical TIL has received relatively little attention, so this is a welcome contribution. The paper includes interesting and novel results and is overall well written and easy to follow. However, I do have some major comments (and a number of minor comments) that should be addressed before publication, see below.

[Figure]

Major Comments:

1) divergence-TIL relationship:

The relation of TIL strength to tropopause-level divergence is new and interesting. But what I find puzzling is that convergence apparently does not lead to a reduction in TIL strength (Fig. 3). For DJF TIL strength is independent of the strength of convergence (Div < 0), for JJA it even increases slightly for strong convergence. This seems to contradict the mechanism put forward in section 3.2 (vertical gradient of vertical velocity forcing Nˆ2) and should be discussed/interpreted somewhere in the paper.

Another question I have related to the divergence-TIL relation is: what is the impact of deep convective outflow? Strong tropopause-level divergence would be expected from organized deep convection. Deep convection is known to be associated with the "cold top" (e.g. Holloway & Neelin, 2007; or Paulik & Birner, 2012 who quantified this using COSMIC data) – a strong tropopause-level cold anomaly aloft mid-to-upper tropospheric heating, which should be associated with enhanced TIL. This signal would primarily show up for strong meso- to large-scale divergence. I wonder whether this in part explains the relationship shown in Fig. 3? For large-scale convergence the TIL may locally still be enhanced due to smaller scale dynamics (e.g. gravity waves) and the tropopause-following coordinate.

2) wave-modulation of tropopause

I found the portrayal of the wave-modulation of the tropopause and TIL somewhat confusing. Section 4 is titled "Dynamical Forcing by Equatorial Waves", but what is primarily shown is the quasi-reversible transient modulation. Any wave with a vertical temperature signature will have layers of positive temperature gradient (∼enhanced stratification) and layers of negative temperature gradient (∼reduced stratification). By definition, if the wave propagates through the tropopause, the tropopause algorithm will place the local tropopause near the wave-induced temperature minimum, which, again by definition, puts the layer of enhanced stratification (∼TIL) just above the local

tropopause. From that perspective, the TIL enhancement is just a quantification of the wave itself, so cannot be considered a response to the wave (as would be implied by "forcing"). It also doesn't allow the TIL to be considered part of the basic state structure for wave propagation (see authors' motivation in 2nd paragraph of abstract and introduction).

I would urge the authors to be more careful with the wording and interpretations in section 4: what is quantified is the wave-modulation of the tropopause (incl. its TIL structure), not the wave-forcing. It is not clear how much of the analyzed signals are reversible vs. irreversible – possibly, a life-cycle analysis of certain wave types might reveal how much of the wave-modulation is left over once the wave has passed through the region.

3) discussion of applicability to extratropics:

I suggest to either expand Section 5 or remove it – it's not much of a discussion at this point, other than to simply note that there are waves in the extratropics and that a similar analysis could be performed there. The way it stands it would suffice to simply mention this in section 6. If the authors feel it's important to include this section then it should discuss in what way the findings might carry over to the extratropics (or not), given the very different dynamical constraints and physically distinct waves. But again, I don't really see the point of including such a discussion – it seems to primarily distract from the main points of the paper.

Minor comments:

Abstract: the first two paragraphs are very general/generic and can probably be condensed into one shorter paragraph.

line 16: do you mean that you approximate the meteorological situation by the 100 hPa divergence field? The divergence field certainly doesn't completely determine the meteorological situation.

line 18: "new feature": I agree that this is quantified better here, but the QBO–static stability relation was already described in Grise et al. (2010), so by itself is not new

line 36: I believe Randel et al. (2007) were the first to demonstrate this from GPS

line 52: Randel et al. (2007) were the first to suggest this mechanism

line 70/71: Grise et al. show a lag-regression of N^2 to QBO index, which includes the entire lower-to-mid stratosphere

line 91: 100 m is the resolution at which the data is provided, which is not the same as the effective physical resolution – please include corresponding remark (see referenced papers on GPS data for details)

line 101: it's -> it is (and similarly at other places)

line 110: I suggest parentheses around (g/theta)

line 125: remove "empty"

line 176: remove "a" before "6.5%"

line 200 (and at other places): usually the n=0 mode is referred to as mixed-Rossby-gravity (MRG) wave (or Yanai wave) – please clarify

line 234 (and other places): referring to the QBO-associated static stability maximum as secondary TIL could be confusing, as it's not always located near the tropopause – I suggest to distinguish those; another potential issue is that Grise et al. already referred to a secondary TIL at the poleward flanks of the inner tropics, which is different from what is referred to as secondary TIL here

line 285 (and other places): I suggest "analogous" instead of "similar" for the comparison between vorticity-TIL and divergence-TIL relations (vorticity and divergence are distinct meteorological fields, so "similar" may be confusing to some readers)

line 302: "absence of Coriolis force" – I don't understand this comment, isn't this just

referring to the continuity Eq., which doesn't depend on the Coriolis force?

line 364 / Figs. 5, 6: why did you decide to show the time-derivatives of T and Nˆ2 (as opposed to just T and Nˆ2)? This came as a surprise to me, so I'd suggest to include a brief statement motivating this choice.

References:

Holloway & Neelin (2007): The convective cold top and quasi equilibrium, J. Atmos. Sci., 64, 1467–1487.

Paulik & Birner (2012): Quantifying the deep convective temperature signal within the tropical tropopause layer (TTL), Atmos. Chem. Phys., 12, 12183–12195.

---

## Author Comment (AC1) · 3 Jun 2016

**Response to Referee #1**

We thank Referee #1 for the helpful comments which helped to improve the manuscript significantly.

In the following, we first explain general changes made in the manuscript, and continue with the point-by-point responses to the reviewer's comments. The referee's comments are in blue font, and our replies are in normal font. Every change made in the revised manuscript is highlighted (please find the highlighted version in the Author Response).

**General comments:**

**New subsection 4.3 and Figure 7**

Motivated by the specific comments 2 and 3 by Joowan Kim in his review, we added subsection 4.3 to the manuscript in order to discuss how much of the TIL is left without the equatorial wave signal, other mechanisms that could enhance the remaining TIL, and the forcing of the secondary $N^2$ maximum. Figure 7 compares the time evolution of the equatorial $N^2$ structure with and without the equatorial wave signal (Thomas Birner asked about this during the SHARP2016 workshop, and we found that making this kind of plot would be the best fit for the purposes of section 4.3).

In Fig. 7 the difference in the TIL region when the equatorial wave signal is subtracted is clear, but the secondary $N^2$ maximum below the descending westerly QBO phase remains the same, and therefore is not directly modulated by Kelvin waves, as we were suggesting in the discussion manuscript version. Since proven untrue, the paragraphs that discussed the forcing of the secondary $N^2$ maximum by the filtered Kelvin waves have been erased (now missing from lines 368, 403, 479 and 563), and now we discuss possible forcings in lines 518-527. We still suggest an indirect effect of Kelvin waves (T signal from wave dissipation), but this cannot be captured by our wavenumber-frequency domain filters once the wave dissipates.

**New Appendix C**

We added a caveat about the filtering of waves with periods of less than 2 days from our daily dataset. Spectral ringing can be an issue with these settings, and could leave a spurious signal in our results (Figure 6), but we checked that the contribution of these periods to the calculated equatorial wave signature of inertia-gravity waves is zero, and therefore doesn't affect our results at all.

**Point-by-point responses to Ref#1 comments**

Major issues

1. I am not an expert on reanalysis data, but as far as I know the quality of reanalysis wind data in the tropics is not as good as one would wish them to be. So the question is: how much can you trust the upper tropospheric horizontal divergence in the tropics? The authors should at least address this issue and try to convince the reader that the quality of the data is sufficient for their purpose.

We agree in that upper-tropospheric winds in ERA-Interim in the tropics are somewhat less accurate than in the extratropics, but we don't think the difference is enough to make tropical 100hPa divergence unreliable for the following reasons:

1) Globally, the performance of ERA-Interim at the 100hPa level is comparable to the operational weather forecast system from ECMWF in terms of root mean squared (RMS) error relative to radiosondes (see Figure 1a from Dee et al., 2011). Note that this figure compares RMS of ERA-Interim 1979 analyses (well before GPS-RO was available), to RMS in operational forecasts in 2007. The 100hPa level in ERA-Interim is as good as one can get from state-of-the-art NWP systems.

2) In the extratropics, the wind difference between in-situ observations and ERA-Interim reanalysis has a 1 standard deviation of about 3m/s for both zonal and meridional winds. In the tropics, this difference at 100hPa is of about 4m/s, meaning that the extratropics have about 75% of the inaccuracy found in tropical upper-tropospheric winds (see Figures 17 and 18 from Poli et al., 2010). Also, the tropical winds at 100hPa don't have the worst performance, since the levels between 120-200hPa in the tropics have a higher 1std difference of 4.5m/s. In addition, the assimilation of GPS-RO observations slightly reduces this differences about everywhere.

3) In-situ observations, radiosondes, have uncertainties as well: several m/s of standard error can be observed applying different tracking techniques, and the errors highly depend on the wind regime, shear and rate of vertical ascent. Also, high-resolution radiosondes include small-scale variations of winds (also up to a few m/s) that cannot be resolved by the model's vertical grid. A thorough description of these issues with wind observations can be found at the "GUIDE TO METEOROLOGICAL INSTRUMENTS AND METHODS OF OBSERVATION" (WMO-No. 8), Part I, chapter 13.

We feel there is no need to discuss this issue in the manuscript, but we added a short sentence in lines 102-104 about it.

2. The authors could clarify the role of tropospheric vertical motion and upper tropospheric horizontal divergence for tropical TIL formation, e.g. in their section 3.2. Assuming that a tropospheric wave produces regional upwelling with horizontal divergence right at the tropopause level, this would yield a higher and sharper than normal tropopause (corresponding to a stronger than normal TIL) — essentially by pushing upward the tropopause and thereby making the lowermost stratosphere somewhat colder. In this simple scenario there is no warming involved at any point: the TIL forms because the cooling has some vertical structure decaying with altitude. On the other hand, composite plots like Figure 6a indicate actually some warming in the lowermost stratosphere. Does this mean that the equatorial waves are associated with downwelling in the lowermost stratosphere (right above the tropospheric upwelling), or does this possibly imply diabatic warming?

Regarding divergence, we connect it to convection and tropopause cooling by the hydrostatic adjustment mechanism. Here the suggestions of Joowan Kim were helpful in providing references for a clearer explanation for the sharper TIL with divergent flow (see specific comment 3 of his review). It has to be noted that this mechanism and our results with divergence from Figures 2 and 3 are independent of equatorial wave activity: deep convection (and near-tropopause divergence) can be coupled to an equatorial wave or not, and is represented either way in the diagram of sTIL versus divergence in Fig. 3. We added a new paragraph discussing this within section 3.2, lines 300-307.

The equatorial wave signature in Figure 6a comes entirely from making a tropopause-based mean of the different wave anomalies: it appears because the tropopause is adjusted to the wave anomalies – a ground based mean gives zero. The reason for this is that a Fast Fourier Transform separates a field into a sum of harmonics, which are deviations from the zonal mean. The constant (ground-based zonal mean) term is not included in the wave signals obtained by the filtering method, and the ground-based sum of the positive-negative parts of each harmonic is zero. It is the tropopause undulations and the tropopause-based averaging that enable the signature in Fig. 6a to appear, and we make this clear throughout section 4.2 now.

Our method is suited to compare the signal of the different equatorial wave types on the TIL: therefore the tropopause-based averaging of the temperature and $N^2$ profiles while creating a gridded dataset, and the tropopause-based averaging of the wave anomalies.

However, conclusions about vertical motion cannot be inferred from Figure 6: the observations we work with are temperatures from GPS-RO and the filtered wave anomalies, and vertical motion is a derived, indirect quantity that can be obtained from models, whose vertical resolution is not enough to enable a study of the relation of upwelling and the small-scale filtered anomalies.

In a scenario of zonal-mean ascent in the upper troposphere, a wave would consist of a harmonic of upwelling and downwelling *anomalies* from this zonal mean*:* there would be a local cooling effect (to which the tropopause would be lifted by the extra upwelling, therefore adjusting to the anomaly), and a local warm anomaly somewhere else, which would fall above the tropopause since it's not necessarily been lifted there. Thus, the tropopause-based zonal mean would show the dipole of tropopause cooling and warming aloft. Once the wave has a vertical phase tilt (a more realistic scenario, e.g. Figure 5a) this dipole can be present in the same place, otherwise the cooling/warming are in different regions. The warm anomaly doesn't imply downwelling per se, it may as well be less upwelling. In this scenario, the existence of the wave doesn't affect the zonal-mean ascent: only the tropopause horizontal structure and the TIL. Non-linear interactions are needed for a wave to change the zonal-mean flow (e.g. wave breaking), these can be complex and are beyond the scope of our study.

Our method and corresponding results in section 4 were specifically designed to target TIL forcing, and they don't give conclusions about vertical motion related to equatorial wave activity. In section 3.2, divergence is related to vertical wind convergence, which doesn't give information about the actual rate of ascent, just its gradient at that level independently of wave activity. For these reasons, we find that a discussion about vertical motion in sections 3.2 and 4 is very difficult to link to our results while not adding insight about the TIL.

1. Line 166: What is an e-fold function? A Gaussian?

We renamed the function into 'exponentially-folding' (l. 169) for better clarity. Note that the mathematical expression of the weighting function is in line 171.

2. Line 172: How are the profiles shifted in altitude? By how much? For what purpose?

The profiles are shifted from a tropopause-based scale onto a ground-based one. We rephrased lines 175-176 to clarify this. The purpose of making tropopause-based averages while gridding GPS-RO profiles is to smooth the TIL as little as possible (l. 190). The filtering has to be done at ground-based levels, since we know the tropopause undulates, adjusts to the equatorial wave signal and is not a constant reference level.

3. If I recall right, an important point in the work of Wheeler and Kiladis (1999) is the removal of the background spectrum. How is this dealt with in the present work?

In Wheeler and Kiladis (1999) the background power spectrum is calculated to discern which regions of the wavenumber-frequency domain have a spectral signature significantly above the background (Fig. 3 of their paper). We don't present such diagrams in our study. While filtering, the inclusion of background noise is unavoidable, but it appears as a continuum of small amplitude fluctuations: please see the beginning of section 4 in Wheeler and Kiladis (1999). The background spectrum doesn't need to be removed since the filtered wave anomalies appear as bursts of high amplitude compared to it.

4. As a standard reference for the seasonal cycle of the tropical tropopause one should add the paper by Yulaeva et al. (1994).

Agreed, this reference was added in line 229.

5. Line 258, ".... temperature inversion is added to this background profile...": For me, "temperature inversion" means that the temperature increases (rather than decreases) with altitude. It seems that this term should only be used for full temperature profiles, not for perturbations or "additions". So I have a difficulty with the expression "adding a temperature inversion to the background profile".

We changed the term 'temperature inversion' for 'dipole of tropopause cooling and warming aloft' in the sentence, see l. 265.

6. Line 259: "skyrocket" appears too colloquial and not quite fitting here.

We changed this term for 'increases dramatically' in the sentence, see l.266.

7. Line 260, "the $N^2_{max}$ is very narrow": strictly speaking this is not true. The peak containing $N^2_{max}$ may be very narrow, not the $N^2_{max}$ itself.

Agreed, the sentence was changed accordingly (see l. 268).

8. Line 336: How is the significance of the difference between the curves assessed? As far as I know, the significance of the difference in the mean between two distributions is measured by the standard error (Press et al., 1992), not by the standard deviation.

The purpose of the sentence was to infer that a significance test is not needed: the two means are separated by 30 standard deviations, which is really far apart. A common way to assess the significance of the difference in the mean of two distributions is a t-test. The difference between the

Easterly-Westerly QBO $N^2_{max}$ distributions is well beyond the 99.9% significance level, as we now explain in lines 356-359. We prefer not to use the term 'standard error' since both distributions are true.

9. Line 364: How was the longitude chosen for the plots in figure 5?

We found that the word 'sections' might have been misleading in the sentence. We changed it for 'snapshots' (l. 380 now). There is no longitude limitation in the plots in Figure 5, note they go from -180 to 180 degE.

10. Line 374, "... tend to be aligned...": Well, this seems to be at least partly wishful thinking, I find that it is sometimes true, but sometimes not.

We rephrased the paragraph so it immediately specifies that the tropopause adjustment happens where the wave anomalies are large (see lines 393-395).

11. Line 378, "... cooling and/or warming...": this is not clear to me.

We changed this expression for 'dipole of TP cooling and warming aloft' while rephrasing lines 393-395. We hope that the paragraph involved in points 10 and 11 is more straightforward now.

12. Line 396 and line 401: Figure 5 shows anomalies of $\delta N^2/\delta t$, not anomalies of T!

Thank you for finding this mistake, the terms were corrected accordingly (now in line 407), and the reference to Fig. 5 in the next paragraph was erased. In lines 411-413 now we refer to our method for clarity, since we do filter both T and $N^2$ fields. Also note that we do not use time derivatives any more in figures 5 and 6, but anomalies (and averaged anomalies), since our earlier interpretation of these quantities was confusing.

13. Line 457, "... a small part...": how do you know that this part is small? Could it be a substantial part?

We erased the term 'small' from this sentence (now in l. 467). We expect the radiative contribution to be small in that equatorial waves are not radiatively driven and their propagation is explained by dry dynamics. We added this explanation in lines 469-471.

14. Line 485, should read: "... would be suited to....".

Thank you for finding this mistake, it's been corrected.

15. Line 525, "... is rather marginal...": "marginal" may not be the right term here. True, it is smaller than in the corresponding figure 3, but it may yet be significant!

We agree. The term 'marginal' was changed for 'very small' (l. 590 and also 298).

**References**

Dee, D. P., and coauthors,(2011): The ERA-Interim reanalysis: configuration and performance of the data assimilation system, Quarterly Journal of the Royal Meteorological Society, 137, 553-597, doi:10.1002/qj.828.

Poli, P., S. B. Healy, and D. P. Dee (2010): Assimilation of Global Positioning System radio occultation data in the ECMWF ERA-Interim reanalysis, Quarterly Journal of the Royal Meteorological Society, 136, 1972-1990, doi:10.1002/qj.722.

GUIDE TO METEOROLOGICAL INSTRUMENTS AND METHODS OF OBSERVATION (WMO-No. 8) PROVISIONAL 2014 EDITION FOR CIMO-16 APPROVAL. Part I, chapter 13: Measurement of upper wind

http://www.wmo.int/pages/prog/www/IMOP/publications/CIMO-Guide/Provis2014Ed/Provisional2014Ed_P-I_Ch-13.pdf

---

## Author Comment (AC2) · 3 Jun 2016

**Response to Referee #2 (Joowan Kim)**

We thank Referee #2 (Joowan Kim) for the helpful comments which helped to improve the manuscript significantly. We are particularly grateful for the suggestion about the hydrostatic adjustment mechanism, which clarifies our results in section 3.2. Also, we included a new section 4.3 to discuss how much of the TIL is left without the equatorial wave signal, other mechanisms that could enhance the remaining TIL, and the forcing of the secondary $N^2$ maximum (as requested in the specific comments). We find that this new subsection puts our results in a better context and makes the paper rounder as a whole.

In the following, we first explain general changes made in the manuscript, and continue with the point-by-point responses to the reviewer's comments. The referee's comments are in blue font, and our replies are in normal font. Every change made in the revised manuscript is highlighted (please find the highlighted version in the Author Response).

**General comments:**

**New subsection 4.3 and Figure 7**

Motivated by the specific comments 2 and 3 by Joowan Kim in his review, we added subsection 4.3 to the manuscript in order to discuss how much of the TIL is left without the equatorial wave signal, other mechanisms that could enhance the remaining TIL, and the forcing of the secondary $N^2$ maximum. Figure 7 compares the time evolution of the equatorial $N^2$ structure with and without the equatorial wave signal (Thomas Birner asked about this during the SHARP2016 workshop, and we found that making this kind of plot would be the best fit for the purposes of section 4.3).

In Fig. 7 the difference in the TIL region when the equatorial wave signal is subtracted is clear, but the secondary $N^2$ maximum below the descending westerly QBO phase remains the same, and therefore is not directly modulated by Kelvin waves, as we were suggesting in the discussion manuscript version. Since proven untrue, the paragraphs that discussed the forcing of the secondary $N^2$ maximum by the filtered Kelvin waves have been erased (now missing from lines 368, 403, 479 and 563), and now we discuss possible forcings in lines 518-527. We still suggest an indirect effect of Kelvin waves (T signal from wave dissipation), but this cannot be captured by our wavenumber-frequency domain filters once the wave dissipates.

**New Appendix C**

We added a caveat about the filtering of waves with periods of less than 2 days from our daily dataset. Spectral ringing can be an issue with these settings, and could leave a spurious signal in our results (Figure 6), but we checked that the contribution of these periods to the calculated equatorial wave signature of inertia-gravity waves is zero, and therefore doesn't affect our results at all.

**Point-by-point responses to Ref#2 (Joowan Kim) comments**

Specific comments (minor)
1. The title is too broad for the contents of the manuscript. Authors are mainly focusing on dynamical mechanisms that could enhance TIL in the tropics. Although they demonstrate the mechanisms clearly, the contents in the manuscript are still too limited to cover the whole spectrum of the tropical TIL (e.g., annual cycle, influence of deep convection and radiation, role of shallow Brewer-Dobson circulation). It is strongly recommend for authors to further specify the title of this manuscript.

We changed the title to "The Tropical Tropopause Inversion Layer: Variability and Modulation by Equatorial Waves", so it's specific to the main results of sections 3 (3D structure and variability) and 4 (effect of equatorial waves).

2. Authors suggest that Kelvin waves cause the enhancement of N2 just below the westerly shear (or zero-wind line) of the QBO. This may be one possible cause, however, the zonal mean temperature anomaly associated with vertical wind shear of the QBO (cf. Fig. 4 in Baldwin et al. 2001) has a strong impact on N2. Several Kelvin of temperature changes in 10 km depth, and this could significantly modulate N2 in the lower stratosphere. In fact, this may have a bigger impact on N2 than Kelvin waves, particularly in zonal mean field. Some analysis and discussion on this effect will be helpful (a simple comparison of tropical mean temperature profiles in westerly and easterly QBO will be good enough).

We prepared a new Figure 7 comparing the $N^2$ structure with and without the equatorial wave signal, in order to see how much of the TIL and the secondary $N^2$ maximum are left without it. We found that the secondary $N^2$ maximum was not affected by the subtraction of equatorial wave anomalies. Therefore, our earlier suggestion that the filtered wave anomalies also contributed to the secondary $N^2$ maximum was found to be wrong, and any reference to this throughout the paper has been erased (now missing from lines 368, 403, 479 and 563).
We still suggest an indirect effect of Kelvin waves (T signal from wave dissipation), but this cannot be captured by our wavenumber-frequency domain filters once the wave dissipates. This and other possible mechanisms forcing the secondary $N^2$ maximum are discussed in lines 518-527. We are thankful for the suggestion about the T anomaly associated to wind shear and the reference regarding this, it was added as well into the discussion.
We hope that the new section 4.3 and Figure 7 fulfill the referee's request in this comment, while compensating the erased discussions about the forcing of the secondary $N^2$ maximum by the filtered equatorial waves.

3. Although influence of deep convection on TIL is beyond the scope of this study, some discussions on tropical convection will still be helpful. For example, the zonal structures in N2 (shown in Fig. 2) are largely related to deep convection in DJF and JJA. In fact, climatology of N2 shows similar structures as in Fig. 2, and this is largely due to tropopause cooling cause by deep convection (deep convection make tropopause colder; e.g., Johnson and Kriete 1982; Gettelman et al. 2002; Paulik and Birner 2012). Only a part of the N2 structure is explained by tropical waves.

In addition, the coherence between N2 and near-tropopause divergence (which is a noble contribution of this paper) is consistent with the hydrostatic adjustment mechanism, which is proposed by Holloway and Neelin (2007) to explain cold-top (tropopause) over deep convection. Those discussion could be helpful for readers.

We added a discussion about other TIL forcing mechanisms in the new section 4.3. Figure 7b shows the remaining, much weaker TIL without the equatorial wave signal. We also include the hydrostatic adjustment mechanism when convection is not wave-coupled and radiative forcing as possible TIL enhancers in this discussion (lines 496-502).

Also, we are very grateful for the suggestion about the hydrostatic adjustment mechanism, regarding the relation between stronger TIL and near-tropopause divergent flow. We added a paragraph in section 3.2 (lines 300-307) discussing this, which improves the explanation about the sTIL relationship with divergence and makes it clearer.

Technical suggestions
Line 35: Satellite GPS => Global Positioning System (GPS) (In many place, satellite GPS => GPS)
We only keep the word 'satellite' the first time we refer to GPS-RO in case any reader is not familiar with this dataset, and we follow the suggestion the rest of the times GPS is referred to.

Line 105: tropopause height (TPz) using the WMO lapse-rate tropopause criterion...
Corrected.

Line 169: latitude (y) and time (t). The maximum distance allowed from the grid point in each dimension is 5°longitude, 10°longitude, and 12 hours, respectively.
Corrected.

Line 214: 3.1 ?
True, there was a title missing! We titled section 3 "Structure and Variability of the Tropical TIL", and subsection 3.1 "Vertical and Horizontal Structures".

Line 234: 2011=>2010?
Thank you for finding this mistake, it was corrected (now in lines 235 and 241).

Line 375: highest amplitude => maximum amplitude
Corrected.

Line 379: very high => very large
Corrected.

Line 393: high amplitude => large amplitude
Corrected.

Line 476: higher that within => larger than that in
Corrected.

Fig 5: why do you show N2 tendency (dN2/dt) instead of N2?
(also in Fig 6: dT/dt instead of T)?

We now show these parameters as anomalies (or averaged anomalies) in both figures 5 and 6. The way we interpreted these quantities was confusing in the earlier manuscript, we hope it is more straightforward now.

---

## Author Comment (AC3) · 3 Jun 2016

**Response to Referee #3**

We thank Referee #3 for the helpful comments which helped to improve the manuscript significantly. We are particularly grateful for the suggestion about the hydrostatic adjustment mechanism, which clarifies our results in section 3.2.

In the following, we first explain general changes made in the manuscript, and continue with the point-by-point responses to the reviewer's comments. The referee's comments are in blue font, and our replies are in normal font. Every change made in the revised manuscript is highlighted (please find the highlighted version in the Author Response).

**General comments:**

New subsection 4.3 and Figure 7

Motivated by the specific comments 2 and 3 by Joowan Kim in his review, we added subsection 4.3 to the manuscript in order to discuss how much of the TIL is left without the equatorial wave signal, other mechanisms that could enhance the remaining TIL, and the forcing of the secondary $N^2$ maximum. Figure 7 compares the time evolution of the equatorial $N^2$ structure with and without the equatorial wave signal (Thomas Birner asked about this during the SHARP2016 workshop, and we found that making this kind of plot would be the best fit for the purposes of section 4.3).

In Fig. 7 the difference in the TIL region when the equatorial wave signal is subtracted is clear, but the secondary $N^2$ maximum below the descending westerly QBO phase remains the same, and therefore is not directly modulated by Kelvin waves, as we were suggesting in the discussion manuscript version. Since proven untrue, the paragraphs that discussed the forcing of the secondary $N^2$ maximum by the filtered Kelvin waves have been erased (now missing from lines 368, 403, 479 and 563), and now we discuss possible forcings in lines 518-527. We still suggest an indirect effect of Kelvin waves (T signal from wave dissipation), but this cannot be captured by our wavenumber-frequency domain filters once the wave dissipates.

New Appendix C

We added a caveat about the filtering of waves with periods of less than 2 days from our daily dataset. Spectral ringing can be an issue with these settings, and could leave a spurious signal in our results (Figure 6), but we checked that the contribution of these periods to the calculated equatorial wave signature of inertia-gravity waves is zero, and therefore doesn't affect our results at all.

**Point-by-point responses to Ref#3 comments**

Major Comments:
1) divergence-TIL relationship:
The relation of TIL strength to tropopause-level divergence is new and interesting. But what I find puzzling is that convergence apparently does not lead to a reduction in TIL strength (Fig. 3). For DJF TIL strength is independent of the strength of convergence (Div < 0), for JJA it even increases slightly for strong convergence. This seems to contradict the mechanism put forward in section 3.2 (vertical gradient of vertical velocity forcing N^2) and should be discussed/interpreted somewhere in the paper.

Another question I have related to the divergence-TIL relation is: what is the impact of deep convective outflow? Strong tropopause-level divergence would be expected from organized deep convection. Deep convection is known to be associated with the "cold top" (e.g. Holloway & Neelin, 2007; or Paulik & Birner, 2012 who quantified this using COSMIC data) – a strong tropopause-level cold anomaly aloft mid-to-upper tropospheric heating, which should be associated with enhanced TIL. This signal would primarily show up for strong meso- to large-scale divergence. I wonder whether this in part explains the relationship shown in Fig. 3? For large-scale convergence the TIL may locally still be enhanced due to smaller scale dynamics (e.g. gravity waves) and the tropopause-following coordinate.

We are grateful for the suggestion about the hydrostatic adjustment mechanism, regarding the relation between stronger TIL and near-tropopause divergent flow from convection. We added a paragraph in section 3.2 (lines 300-307) discussing this, which improves the explanation about the sTIL relationship with divergence and makes it clearer. We link the vertical wind convergence term to convection as well, since it only has an effect on the TIL with divergent flow (convective outflow). We now suggest that vertical wind convergence is one mechanism enhancing the TIL at all latitudes, but caused by different processes: convection in the tropics and baroclinic waves in the extratropics. This interpretation is added in lines 317-322.

2) wave-modulation of tropopause
I found the portrayal of the wave-modulation of the tropopause and TIL somewhat confusing. Section 4 is titled "Dynamical Forcing by Equatorial Waves", but what is primarily shown is the quasi-reversible transient modulation. Any wave with a vertical temperature signature will have layers of positive temperature gradient (enhanced stratification) and layers of negative temperature gradient (reduced stratification). By definition, if the wave propagates through the tropopause, the tropopause algorithm will place the local tropopause near the wave-induced temperature minimum, which, again by definition, puts the layer of enhanced stratification (TIL) just above the local tropopause. From that perspective, the TIL enhancement is just a quantification of the wave itself, so cannot be considered a response to the wave (as would be implied by "forcing"). It also doesn't allow the TIL to be considered part of the basic state structure for wave propagation (see authors' motivation in 2nd paragraph of abstract and introduction).

I would urge the authors to be more careful with the wording and interpretations in section 4: what is quantified is the wave-modulation of the tropopause (incl. its TIL structure), not the wave-forcing. It is not clear how much of the analyzed signals are reversible vs. irreversible – possibly, a life-cycle analysis of certain wave types might reveal how much of the wave-modulation is left over once the wave has passed through the region.

We agree that the use of the term 'forcing' might not be the most correct, it's a good point that it shall rather be considered as an instantaneous modulation. We substituted the term 'forcing' for 'modulation' throughout the paper, also for 'signal' or 'signature' where it was most convenient. We discuss the possibility of further (more permanent) effects of the waves once they leave the tropopause (or dissipate), which could enhance the TIL as well, in lines 513-517.

Regarding the wave signature as a mere quantification of the wave itself, it shall not be viewed as an artifact of the tropopause-coordinate following. Although transient and instantaneous, there are motions associated to the wave signal that locally lift/cool/modulate the tropopause, and also warm the air aloft. Another characteristic of the waves is that they amplify next to and above the tropopause (Fig. 5), and also increase their vertical tilt (Fig. 5a, visible for Kelvin waves), which increases the wave signal in the TIL region, and also increases the area of positive $N^2$ anomaly above the tropopause. This is a response of the wave to the elevated $N^2$ values in the lowermost stratosphere, in agreement with linear theory, which in turn enhances the TIL further, working as a TIL-enhancing feedback. We discuss this in lines 480-489, while specifying that more research needs to be done to ascertain such feedback as a robust feature of the tropopause region.

We prefer not to discuss whether the TIL shall be considered part of the basic state structure for wave propagation in our manuscript since it's beyond the scope of our study, and our current results are not enough to fully support (or deny) this. Nevertheless, our results suggest that there is a response of the wave to the higher $N^2$ values in the lowermost stratosphere (Fig. 5a), which is predicted by theory. But again more research needs to be done in this respect in order to make a robust statement.

3) discussion of applicability to extratropics:
I suggest to either expand Section 5 or remove it – it's not much of a discussion at this point, other than to simply note that there are waves in the extratropics and that a similar analysis could be performed there. The way it stands it would suffice to simply mention this in section 6. If the authors feel it's important to include this section then it should discuss in what way the findings might carry over to the extratropics (or not), given the very different dynamical constraints and physically distinct waves. But again, I don't really see the point of including such a discussion – it seems to primarily distract from the main points of the paper.

Since the submission of the manuscript, there have been developments regarding the application of our method in the extratropics: the method is successful in quantifying the modulation and enhancement of the TIL in the extratropics by extratropical waves, and we are preparing an upcoming paper about this.

We would prefer to keep this section (with some rephrasing in lines 544-548), since it is important to state the usefulness of our method outside the equator. Also note that we do discuss about what is to be expected in the extratropics: see the discussion about baroclinic Rossby waves in lines 531-538 (which are expected to have a bigger signature than Kelvin or any equatorial wave). The role of inertia-gravity waves in enhancing the extratropical TIL is predicted from the modelling study by Kunkel et al. (2014), but is still awaiting confirmation from observations.

Minor comments:

Abstract: the first two paragraphs are very general/generic and can probably be condensed into one shorter paragraph.

We merged both paragraphs into one, slightly reducing its length where possible.

line 16: do you mean that you approximate the meteorological situation by the 100 hPa divergence field? The divergence field certainly doesn't completely determine the meteorological situation.

We agree in that we do not show a full meteorological description of the tropical tropopause, but only TIL-relevant parameters. We erased the word 'meteorological' to make the sentence simpler.

line 18: "new feature": I agree that this is quantified better here, but the QBO–static stability relation was already described in Grise et al. (2010), so by itself is not new

We agree that Grise et al. (2010) shows a correlation of enhanced $N^2$ in the layer 1-3 above the tropopause and throughout the lower-mid stratosphere following the easterly phase of the QBO. However, in that paper there is no reference that this correlation creates a second $N^2$ maximum that is close to TIL strength. We feel that it is justified to call it a new feature.

line 36: I believe Randel et al. (2007) were the first to demonstrate this from GPS

In the paper by Randel et al. (2007) the term 'global' is used in a very nuanced way: they show the "global structure of the _extratropical_ TIL". The tropics and the equator are not investigated in this paper. Grise et al. (2010) is the first publication to explore the TIL globally in the literal sense: covering the extratropics and tropics, therefore we would like to keep the reference as is.

line 52: Randel et al. (2007) were the first to suggest this mechanism

We agree, we added this reference in the text.

line 70/71: Grise et al. show a lag-regression of N^2 to QBO index, which includes the entire lower-to-mid stratosphere

We agree and we refer to this later in section 3.1.1, but we feel that including this into a TIL-relevant introduction is unnecessary.

line 91: 100 m is the resolution at which the data is provided, which is not the same as the effective physical resolution – please include corresponding remark (see referenced papers on GPS data for details)

We added the corresponding remark in lines 90-91.

line 101: it's -> it is (and similarly at other places)

We corrected this throughout the paper.

line 110: I suggest parentheses around (g/theta)

We added parentheses for both terms in the equation.

line 125: remove "empty"
      Corrected.

line 176: remove "a" before "6.5%"
      Corrected.

line 200 (and at other places): usually the n=0 mode is referred to as mixed-Rossby-gravity (MRG) wave (or Yanai wave) – please clarify
      We added a clarification in lines 206-207: we don't use these terms for simplicity, since there's already a considerable array of wave types that we filter and constantly refer to throughout the paper.

line 234 (and other places): referring to the QBO-associated static stability maximum as secondary TIL could be confusing, as it's not always located near the tropopause – I suggest to distinguish those; another potential issue is that Grise et al. already referred to a secondary TIL at the poleward flanks of the inner tropics, which is different from what is referred to as secondary TIL here
      We added a specification in lines 239-240 that this secondary maximum shall not be considered a second TIL. Note that throughout the paper we differentiate between TIL and this secondary maximum (see line 251, 335, 363), and that this secondary $N^2$ maximum is never referred to as a second TIL, only that it leads to a double-TIL-like structure in static stability (because it looks like it in the stability profile, but the second maximum is far away from the tropopause, and strictly speaking there is no temperature inversion given the background temperature lapse-rate in the stratosphere).
      In the paper by Grise et al. (2010) there is reference to two distinct features in static stability (in the layers 0-1 and 1-3 km above the tropopause), not to a secondary TIL.

line 285 (and other places): I suggest "analogous" instead of "similar" for the comparison between vorticity-TIL and divergence-TIL relations (vorticity and divergence are distinct meteorological fields, so "similar" may be confusing to some readers)
      Thank you for this suggestion, we proceeded with this change throughout the paper.

line 302: "absence of Coriolis force" – I don't understand this comment, isn't this just referring to the continuity Eq., which doesn't depend on the Coriolis force?
      We now see that this sentence was misleading in the way it was written: we wanted to imply that in the tropics the vertical convergence term is not related to relative vorticity. We erased that part of the sentence for simplicity (line 317 now), since parts of section 3.2 have been rephrased and the separation of the processes driving vertical convergence in the tropics/extratropics are clear now.

line 364 / Figs. 5, 6: why did you decide to show the time-derivatives of T and N^2 (as opposed to just T and N^2)? This came as a surprise to me, so I'd suggest to include a brief statement motivating this choice.
      We now show these parameters as anomalies (or averaged anomalies) in both figures 5 and 6. The way we interpreted these quantities was confusing in the earlier manuscript, we hope it is more straightforward now.

---

## Referee Report (RR1)

**Review of the resivsed manuscript** "The tropical tropopause inversion layer: variability and modulation by equatorial waves " by R. P. Kedzierski, K. Matthes and K. Bumke, submitted for publication to ACP

The authors have thoroughly revised their manusript. Overall this lead to a significant improvement; in particular, one misinterpretation was detected (by one of the other reviewers) and rectified in the revised manuscript. The authors have addressed my original concerns in some way.

Having said this, I suggest that the authors still could put in some effort in order to improve the text. Below are my concerns and suggestions. Since some of my remarks are new, I leave it up to the authors and/or the editor to decide to what extent these suggestions should be taken in to accout upon revision.

1. A key concept in this paper is "the divergence". The authors should at some point state clearly that they refer to the divergence of the horizontal wind field in this context. To what extent is this "divergence" different from "vertical wind convergence", which is used somewhere else in the text?

2. In my original review I formulated some concern regarding the suitability/accuracy of reanalysis data for the purpose of the paper. I did not find the authors' reply completely satisfying. At issue is not the typical magnitude of the absolute wind error, but rather how accurately the divergence of the upper troposheric horizontal wind field can be calculated. In the tropics this divergence can be expected to be ralated to some extent to convection and tropospheric diabatic heating, and to my knowledge the forecast models used to be somewhat deficient in this respect.

   I suspect that this is not a big issue on the (rather large) spatial scales which the authors consider, and this is also suggested by the consistency of the results. Yet, the authors could provide a short discussion somewhat more to this point, because the usability of the reanalysis data is very important for the results of this paper.

3. I am not really happy yet with the use of the terms "warming" and "cooling". The authors typically consider band-with filtered wave signals, in other words anomalies from the zonal mean. If a plot shows local warm or cold anomalies, this does not necessarily imply that there is/was warming or cooling. It could just as well be the result of horizontal advection, i.e. that the original air was replaced by warmer or colder air, and this would not imply any warming or cooling (neither diabatic nor adiabatic). Using more precise terminology could make the discussion of the processes more lucid.

4. Following on the previous item, the authors should at some point define what $\Delta$ means, e.g. in the axis labels of their figure 6; similarly for the use of the term "anomaly", e.g. in line 419; similarly $\Delta N^2$ in the color bar of figure 5.

**Minor issues**

1. Is "expontially folding function" really a good terminology in order to refer to a Gaussian-shape function?

2. I would like to see a short phrase explaining the "hydrostatic adjustment mechanism" (line 301) such that the reader can understand this mechanism without consulting the reference Holloway and Neelin (2007).

3. Line 333: "correlation of enhanced $N^2$....": ... with what other variable? (Usually one correlates variable A with variable B)

4. In the paragraph on lines 452–457 the authors talk about heating and cooling rates (K/day) and rates of change of $N^2$. On the other hand, the figures they discuss in this paragraph (figures 5 and 6) show anomalies $\Delta$ owing to the wave. How does this go together?

5. I do not find the statement on line 470 very logical: true, tropical waves may primarily be of dynamic origin, but does this necessarily imply that the radiative signal is small? ... small in what sense?

6. Line 598: should read "As shown....".

---

## Author Response (AR2)

**Response to reviews of Pilch Kedzierski et al.: "The Tropical Tropopause Inversion Layer: Variability and Modulation by Equatorial Waves"**

Dear editor,

  We would like to thank the reviewers for their helpful and encouraging comments in this second iteration. We are particularly grateful for the notes about GPS-RO effective resolution, as well as the comments about the interpretation of the filtered wave anomalies throughout section 4. These needed another update for more clarity and precision of the text.
  In the following paragraphs we include our point-by-point response to each comment in the reviews along with the changes made in the manuscript. The referee's comments are in blue font, and our replies are in normal font. Every change made in the revised manuscript is highlighted. We hope that the new version fulfills the reviewer's requests.
  Yours sincerely,

Robin Pilch Kedzierski
Katja Matthes
Karl Bumke

**Point-by-point responses to Ref#1**

1. A key concept in this paper is "the divergence". The authors should at some point state clearly that they refer to the divergence of the horizontal wind field in this context. To what extent is this "divergence" different from "vertical wind convergence", which is used somewhere else in the text?
  It is true that we did not specify divergence to be from horizontal winds throughout the manuscript. We make this clearer now in lines 14, 100, 133, 258, 278, 289, 565; especially at the beginning of each section that refers to divergence. We also specify this now in the captions of Figures 2 and 3.
  Note that we relate horizontally divergent flow to vertical convergence for continuity reasons in lines 323-324.

2. In my original review I formulated some concern regarding the suitability/accuracy of reanalysis data for the purpose of the paper. I did not find the authors' reply completely satisfying. At issue is not the typical magnitude of the absolute wind error, but rather how accurately the divergence of the upper troposheric horizontal wind field can be calculated. In the tropics this divergence can be expected to be ralated to some extent to convection and tropospheric diabatic heating, and to my knowledge the forecast models used to be somewhat deficient in this respect. I suspect that this is not a big issue on the (rather large) spatial scales which the authors consider, and this is also suggested by the consistency of the results.

Yet, the authors could provide a short discussion somewhat more to this point,
because the usability of the reanalysis data is very important for the results of
this paper.

We briefly extended the short discussion about the accuracy of ERA-Interim horizontal wind divergence, now in lines 104-107. We agree that this not a big issue: even if biases in the model are present, the synoptic-scale variability of divergence shall be in balance with the temperature variability which is constrained by the GPS-RO data that ERA-Interim assimilates, and smaller scales that are not resolved by GPS-RO are not investigated in our study.

3. I am not really happy yet with the use of the terms "warming" and "cooling".
The authors typically consider band-with filtered wave signals, in other words
anomalies from the zonal mean. If a plot shows local warm or cold anomalies,
this does not necessarily imply that there is/was warming or cooling. It could
just as well be the result of horizontal advection, i.e. that the original air was
replaced by warmer or colder air, and this would not imply any warming or
cooling (neither diabatic nor adiabatic). Using more precise terminology could
make the discussion of the processes more lucid.

We substituted the terms 'warming' and 'cooling' by 'cold/warm anomalies' throughout the paper. See lines 25, 430-431, 438, 448-449, 463, 489, 584-585. In the case where we use cooling/warming we refer to the net change of the tropopause-based profile, not to cooling/warming of the air (l. 593).

4. Following on the previous item, the authors should at some point define what
Δ means, e.g. in the axis labels of their figure 6; similarly for the use of the term
"anomaly", e.g. in line 419; similarly ΔN2 in the color bar of figure 5.

We realized that some changes made in the figure labels in the previous iteration were not completely updated at the beginning of section 4.1 in the text, as well as in the figure captions. The use of the terms 'anomaly' and the averaged anomalies has been clarified now in section 4 and the figure captions.

Minor issues
1. Is "expontitly folding function" really a good terminology in order to refer to a
Gaussian-shape function?

Although 'e-fold' function was used in Randel and Wu 2005 and we followed this terminology, we agree that 'Gaussian-shape' is the most precise term for the mathematical formulation of the weights, which is now updated in line 172.

2. I would like to see a short phrase explaining the "hydrostatic adjustment mech-
anism" (line 301) such that the reader can understand this mechanism without
consulting the reference Holloway and Neelin (2007).

We included a short explanation of this mechanism in lines 305-309.

3. Line 333: "correlation of enhanced N2....": ... with what other variable? (Usually one correlates variable A with variable B)

We rephrased this sentence to follow this suggestion, now in line 341.

4. In the paragraph on lines 452-457 the authors talk about heating and cooling rates (K/day) and rates of change of N2. On the other hand, the figures they discuss in this paragraph (figures 5 and 6) show anomalies Δ owing to the wave. How does this go together?

We realized that some changes made in the figure labels in the previous iteration were not completely updated in the text in section 4.2, regarding cooling/heating rates. We substituted these terms for 'mean cold / warm anomalies' or '…K colder tropopause'. We also erased the unit '/day' in the text in section 4.2 since it does not apply for the updated interpretation, so it is not referred to as a cooling/heating rate any longer; only as average anomalies in the seasonal zonal-mean tropopause-based profiles.

5. I do not find the statement on line 470 very logical: true, tropical waves may primarily be of dynamic origin, but does this necessarily imply that the radiative signal is small? ... small in what sense?

Near-tropopause height cloud tops are not frequent, especially the convective type, which constrains the relative contribution of radiation to the total wave signal near the tropopause. We added this into the sentence in line 480.

We also added a caveat in the next paragraph (l. 482-485) about the inclusion of the hydrostatic adjustment from deep convection into the wave signal, but this is also a dynamical response.

Corrected.

**Point-by-point responses to Ref#2 (Joowan Kim) comments**

Quick comment:
Line 90: physical resolution of GPS RO is lower than 100 m in most of the upper troposphere and stratosphere. Authors may want to remove the sentence "(although the effective physical … better)".

Thank you for finding that this sentence was incorrect, this was quite a mistake. It is true that currently the effective vertical resolution of GPS-RO is between ~0.1-1km, better where stratification changes, e.g. the top of the boundary layer or the tropopause region. Although the theoretical limit for GPS-RO could be ~60m, this is not yet implemented!

We rephrased lines 89-93 to clarify this.

**Point-by-point responses to Ref#3**

line 91: this is incorrect - the effective resolution is not as good as the 100 m that the data is provided in, it's more like ~1 km (but tends to be better at places of near-discontinuous stratification such as the tropopause)

Thank you for finding that this sentence was incorrect, this was quite a mistake. It is true that currently the effective vertical resolution of GPS-RO is between ~0.1-1km, better where stratification changes, e.g. the top of the boundary layer or the tropopause region. Although the theoretical limit for GPS-RO could be ~60m, this is not yet implemented!

We rephrased lines 89-93 to clarify this.

line 206: I suggest modifying this statement to say something like "Our WIG category includes Mixed-Rossby-Gravity modes for n=0." The statement as is might imply that you suggest the MRG mode could be interpreted as WIG mode, which is wrong.

We followed this suggestion.

line 506: perhaps this discussion could also mention the MJO, which is associated with organized deep convection and likely falls at least partly into the "hydrostatic adjustment" category

Note that if convection is coupled to an equatorial wave, it will travel in the same wavenumber-frequency domain as the wave, therefore being captured by the filters. In the case of the MJO, the associated signal from convection and the hydrostatic adjustment would be included within the filtered anomalies. We included a caveat about the inclusion of the hydrostatic adjustment mechanism into the filtered anomalies in lines 482-485. The hydrostatic adjustment signal from convection that is not coupled to any equatorial wave can be responsible for the structures observed in Fig. 7b as discussed in section 4.3, and this excludes the MJO since we filter it out.

In the same way in section 3.2, the relation of stronger TIL with divergent flow does not differentiate convection coupled to equatorial waves or not. In both cases, the methodology needs significant refinement to separate the relative contributions from equatorial waves and convection alone.

[revised manuscript text omitted]